# Regulation of chitinase-3-like-1 in T cell elicits Th1 and cytotoxic responses to inhibit lung metastasis

Do-Hyun Kim[1,2], Hong-Jai Park[3], Sangho Lim[1,2], Ja-Hyun Koo[1,2], Hong-Gyun Lee[1,2], Jin Ouk Choi[1], Ji Hoon Oh[4], Sang-Jun Ha[4], Min-Jong Kang [5], Chang-Min Lee[6], Chun Geun Lee [6,7], Jack A. Elias[6,8] & Je-Min Choi[1,2,9]

Chitinase-3-like-1 (Chi3l1) is known to play a significant role in the pathogenesis of Type 2 inflammation and cancer. However, the function of Chi3l1 in T cell and its clinical implications are largely unknown. Here we show that Chi3l1 expression was increased in activated T cells, especially in Th2 cells. In addition, Chi3l1-deficient T cells are hyper-responsive to TcR stimulation and are prone to differentiating into Th1 cells. Chi3l1-deficient Th1 cells show increased expression of anti-tumor immunity genes and decreased Th1 negative regulators. Deletion of Chi3l1 in T cells in mice show reduced melanoma lung metastasis with increased IFNγ and TNFα-producing T cells in the lung. Furthermore, silencing of Chi3l1 expression in the lung using peptide-siRNA complex (dNP2-siChi3l1) efficiently inhibit lung metastasis with enhanced Th1 and CTL responses. Collectively, this study demonstrates Chi3l1 is a regulator of Th1 and CTL which could be a therapeutic target to enhance anti-tumor immunity.

[1] Department of Life Science, College of Natural Sciences, Hanyang University, Seoul 04763, Korea. [2] Research Institute for Natural Sciences, Hanyang University, Seoul 04763, Korea. [3] Department of Internal Medicine, Yale University School of Medicine, New Haven, CT 06510, USA. [4] Department of Biochemistry, College of Life Science and Biotechnology, Yonsei University, Seoul 03722, Korea. [5] Section of Pulmonary, Critical Care and Sleep Medicine, Yale University School of Medicine, New Haven, CT 06510, USA. [6] Department of Molecular Microbiology and Immunology, Brown University, Providence, RI 02912, USA. [7] Department of Internal Medicine, Hanyang University College of Medicine, Seoul 04763, Korea. [8] Division of Medical and Biological Sciences, Warren Alpert Medical School, Brown University, Providence, RI 02903, USA. [9] Center for Neuroscience Imaging Research (CNIR), Institute for Basic Science (IBS), Suwon 16419, Korea. Correspondence and requests for materials should be addressed to J.-M.C. (email: jeminchoi@hanyang.ac.kr)

Chitinase is a defensive enzyme in plant to cleaves chitin and protects hosts against pathogens[1–3]. Chitinase-like proteins (CLPs) are evolutionarily conserved in mammals but do not have the enzymatic activity to directly degrade chitin[4,5]. They have evolved to be important in the development and progression of Th2 inflammation, parasitic infections, and cancer[6–11].

Ym-1, also called chitinase 3-like 3 (Chi3l3), and Ym-2, also called chitinase 3-like 4 (Chi3l4), are CLPs that are important contributors to Th2 inflammation in allergies[7]. A previous study has demonstrated that Ym-1 and Ym-2 could increase the number of γδ T cells and IL-17 production in a nematode infection model[9]. This result demonstrates their evolutionarily conserved roles as innate defense systems. Chitinase 3-like 1 (Chi3l1), also known as breast regression protein 39 (BRP-39), has been more emphasized in cancer and lung inflammation due to human homolog YKL-40, which is mainly expressed in breast cancer cells and lung macrophages[6,10–12]. In *Streptococcus pneumoniae* infections, Chi3l1 regulates macrophage cell death to promote bacterial clearance, indicating a function in innate immunity against pathogens[13]. Studies of transgenic mouse models show that both BRP-39 and YKL-40 are critical regulators of Th2 inflammation in the lung. OVA sensitization induces Chi3l1 expression by macrophages, and induced asthmatic Th2 inflammation is markedly diminished in BRP-39-deficient background, while lung specific YKL-40 overexpression rescues the Th2 inflammation[6]. In addition, Chi3l1 deficiency significantly ameliorates IL-13-induced fibrosis and IL-18-mediated IL-13 production, suggesting an essential role of Chi3l1 in the pathogenesis of Th2 inflammation[6,14].

Chi3l1 was recently found to be involved in pulmonary metastasis[10,11]. The expression of Chi3l1 was significantly increased by melanoma tumor cell challenge and Chi3l1 deficiency reduced lung metastasis of melanoma or breast cancer cells. Although studies have shown that Th2-driven inflammation in the lung induces BRP-39 and it contributes to Th2 inflammation, fibrosis, and tumors, the direct function of Chi3l1 in adaptive immunity such as T cell responses is totally unknown.

Th2 polarization and IL-4 have been shown to promote tumor growth and metastasis[15,16]. Therefore, we hypothesized that specific immune regulatory function of Chi3l1 that drive Th2 polarization while inhibiting Th1 activation may contribute to the tumor development and progression. Here we demonstrated that Chi3l1 expression was highly induced in activated T cells and Th2 cells. Chi3l1-deficient T cells were differentiated into T cells with Th1-prone phenotypes with hyper-responsive to IFNγ signaling and melanoma lung metastasis was significantly reduced in the mice with both Chi3l1 total knock out and CD4⁻Cre system. In addition, in vivo siRNA silencing of Chi3l1 with a cell-penetrating peptide dNP2 efficiently inhibited melanoma lung metastasis by increasing both Th1 and cytotoxic T-lymphocyte (CTL) responses. When viewed in combination, these studies suggest that Chi3l1 plays an essential role in regulation of Th1 and CTL differentiation. These studies highlight that specific intervention of Chi3l1 in T cells could be an effective therapy of pulmonary metastasis.

## Results

**Chi3l1 negatively regulates T cell activation.** Previous studies have reported that CLPs are highly expressed in lung tissue, especially macrophages[6,17]. However, expression of CLPs in lymphocytes, especially in T cells, has not been studied. We performed comparative analysis of mRNA levels of Chi3l1, Chitotriosidase, AMCase, and Ym-1 in splenic macrophages, DCs, T cells, B cells, and NK cells (Supplementary Fig. 1). These

chitinases and CLPs were highly expressed in macrophages however the expression of Chi3l1 was most prominent among these chitinases and CLPs in CD4 and CD8 T cells. In addition, we examined chitinase and CLP expression in naïve and activated CD4 and CD8 T cells. Both mRNA (Fig. 1a, b) and protein (Fig. 1c, d) level of Chi3l1 was strongly induced time dependently upon anti-CD3 and anti-CD28 stimulation. In addition, Chi3l1 expression was strongly induced in Th0 and Th2 cells compare to the other effector T cell subsets (Fig. 1e), suggesting a potential role of Chi3l1 during T cell activation and differentiation. On the other hand, no significant abnormalities were noted in immune cells development between WT and Chi3l1 knock out (KO) mice (Supplementary Fig. 2). To investigate whether Chi3l1 is a negative regulator of T cell activation, MACS-purified naïve CD4 and FACS-purified CD8 T cells from wild type (WT) and Chi3l1 KO mouse splenocytes were activated by anti-CD3 and anti-CD28 antibodies for 3 days. Cytokine flow cytometric analysis demonstrated that Chi3l1 deficiency in both CD4 and CD8 T cells significantly increased IFNγ production, while IL-4 production decreased (Fig. 2a, b). ELISA analysis of Chi3l1 KO CD4 T cells showed increase of both IFNγ and IL-2, decrease of IL-4, and no difference was seen in IL-17 (Fig. 2c) suggesting Th1-like character. In addition, there was increased IFNγ and TNFα expression in Chi3l1 KO CD8 T cells (Fig. 2d), suggesting that Chi3l1 negatively regulates T cell activation, especially regulates Th1 or CTL cytokine production. Because IL-2 and IFNγ are important cytokines for T cell proliferation, we confirmed the proliferation properties of Chi3l1 KO T cells compared to wild type. CFSE-labeled Chi3l1 KO CD4 T cells showed a higher proportion of divided cells in response to TcR stimuli compared to control T cells (Fig. 2e, f). In addition, phosphorylation of Akt, Erk, STAT1, and STAT5 was increased in Chi3l1 KO T cells, suggesting that Chi3l1 is an important factor for regulating T cell proliferation (Fig. 2g, h). Collectively, these results demonstrate that Chi3l1 is expressed in T cells and regulate T cell activation and proliferation for controlling effector functions.

**Chi3l1 inhibits Th1 differentiation via IFNγ-STAT1 axis.** Because Chi3l1 expression was the most significant in Th2 cells and its deficiency in T cells led to increased IL-2 and IFNγ, but decreased IL-4, we hypothesized that it regulates effector T cell differentiation between Th1 and Th2. To investigate whether Chi3l1 regulates Th1 and Th2 differentiation, MACS-sorted CD4⁺CD62L^high CD44^low WT and Chi3l1 KO naïve CD4 T cells were differentiated into Th1, 2, and Th17 cells under specific skewing conditions. The proportion of IFNγ-producing cells significantly increased not only in Th1 but also in Th2 and Th17 conditions while IL-4 in Th2 was decreased and showed no difference in Th17 (Fig. 3a, b). Accumulated cytokines in supernatants showed equivalent patterns (Fig. 3c). T-bet, Runx3, and Eomes, which are important transcription factors of Th1 differentiation, increased in Chi3l1-deficient Th1 cells with increased IFNγ and GM-CSF. JunB, which is important for IL-4-related Th2 differentiation, was significantly reduced, and no differences were observed in Th17-related factors (Fig. 3d). In Chi3l1-deficient Th1 cells, the level of phosphorylated STAT1 was significantly increased, with no difference in the level of phosphorylated STAT4 suggesting Chi3l1 regulates IFNγ signaling not IL-12 (Fig. 4a). On the other hand, phosphorylated STAT6 level was significantly reduced in Th2, collectively suggesting that Chi3l1 could be a negative regulator of IFNγ signaling to commit T cells to more like Th2 cells. In addition, the proportion of IFNγ-producing cells was significantly decreased by IFNγ-neutralizing antibody in Chi3l1 KO Th1 cells (Fig. 4b, c). At 20 µg/mL neutralizing antibody, IFNγ-producing cells were equivalent in WT and Chi3l1 KO Th1 cells. We also

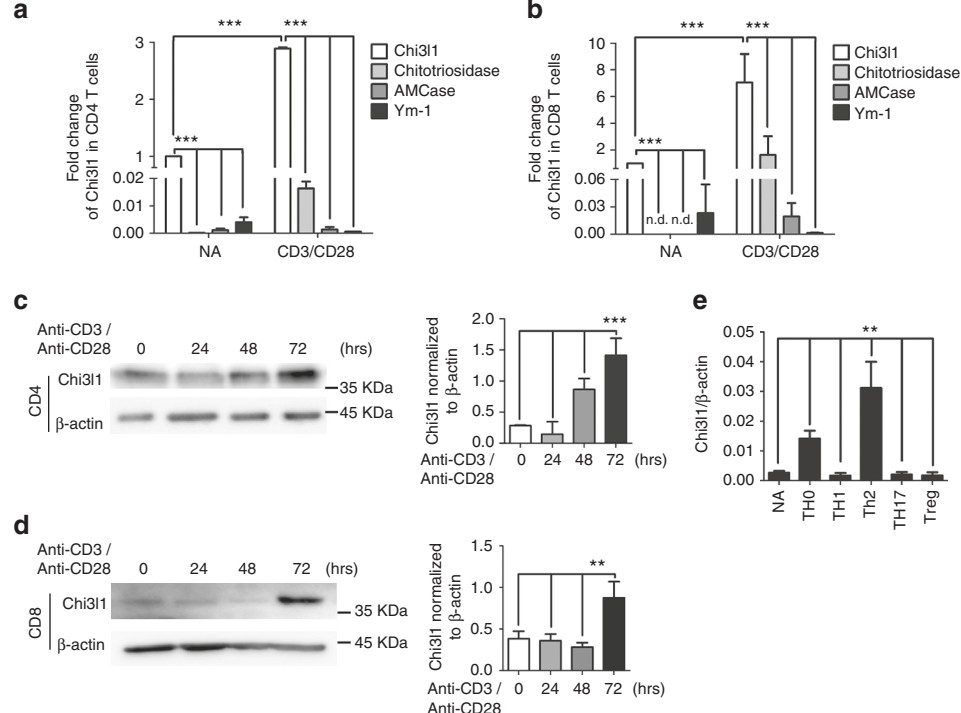

**Fig. 1** Chitinase and Chitinase-like protein expressions in T cells. **a**, **b** mRNA expression of chitinase (Chitotriosidase, AMCase) and chitinase-like protein (Chi3l1, Ym-1) in mouse CD4 and CD8 T cells upon anti-CD3 and anti-CD28 stimulation for 3 days. Each gene expression level was normalized to β-actin, and represented as fold change to Non-activated (NA). **c**, **d** Chi3l1 protein level in CD4 and CD8 T cells stimulated by plate-bound anti-CD3 and anti-CD28 antibodies for indicated time point. Densitometric values of band intensity were calculated by normalization to the value of β-actin. **e** Chi3l1 mRNA expression in naïve and effector T cell subsets. Chi3l1 mRNA expression level was normalized to β-actin level. Data are mean ± SD of three sets of independent experiments ($n = 6$). n.d., not detected; **$p < 0.01$, ***$p < 0.001$ (two-tailed Student's t-test)

treated recombinant IFNγ cytokine to WT and Chi3l1 KO naïve CD4 T cells that Chi3l1 KO T cells showed significantly increased pSTAT1 level compared to WT T cells (Fig. 4d) suggests that increased IFNγ production by Chi3l1 deficiency in T cells could be dependent on increased IFNγ-STAT1 signaling. In addition, we confirmed IFNγ dependency in CD8 T cells, and found that IFNγ and Granzyme B producing activated Chi3l1 KO CD8 T cells were decreased by neutralizing IFNγ antibody treatment, and the level was equivalent with that in WT T cells at 20 μg/mL IFNγ neutralizing antibody (Fig. 4e, f). Taken together, these results demonstrate that Chi3l1 negatively regulates Th1 or CD8 effector T cell differentiation by inhibiting IFNγ-mediated STAT1 phosphorylation.

**Chi3l1 suppresses Th1-related and tumoricidal genes**. To investigate how Chi3l1 regulates the IFNγ-mediated signaling pathway, we isolated RNA from WT and Chi3l1 KO naïve CD4 T cells and differentiated Th1 cells. The transcriptomes of WT and Chi3l1 KO, naïve, and Th1 cells were analyzed by RNA sequencing. To determine the differences in gene expression between WT and Chi3l1 KO Th1 cells, we classified genes that were highly expressed in Th1 cells compared with naïve T cells. We also classified genes that were differently expressed in Chi3l1 KO Th1 cells compared to WT Th1 cells (Fig. 5a). We found 31 up-regulated genes and 72 down-regulated genes in Chi3l1 KO Th1 cells compared to WT Th1 cells. A heatmap obtained by clustering methods showed decreased twist1 and socs1, which are IFNγ-regulatory genes, and increased ifng, ctse, cxcr2, and tnfsf10 (as known as TRAIL), which are tumoricidal molecules, in Chi3l1 KO Th1 cells (Fig. 5b). A scatterplot summarized these patterns (Fig. 5c). To confirm the RNA-sequencing results, we performed quantitative RT-PCR analysis of WT and Chi3l1 KO Th1 cells

(Fig. 5d). Expression of SOCS1, a phosphatase that inhibits phosphorylation of STAT1, was significantly decreased without differences in SOCS3 and SOCS5 in Chi3l1-deficient Th1 cells compared to WT Th1 cells. Twist1, a Th1 inhibitory molecule, was also reduced, presumably explaining how Chi3l1 KO Th1 cells produce more CTSE, TRAIL, IFNγ, and CXCR2. The molecular expression patterns were the same in CD8 T cells (Fig. 5e). In addition, activated Chi3l1 KO CD8 T cells expressed higher levels of T-bet, IFNγ, Perforin, and Granzyme B, suggesting that Chi3l1 KO CD8 T cells induce high levels of effector molecules related to CTL function. Taken together, these results suggest Chi3l1 is a negative regulator of gene expressions regarding Th1 and CTL functions.

**Chi3l1 deficiency in T cells inhibits pulmonary metastasis**. The increase of effector functional molecules in Chi3l1-deficient T cells prompted us to investigate whether Chi3l1-deficient Th1 and CTL contributed to anti-tumor immunity. To test our hypothesis, we utilized B16F10 melanoma lung metastasis model. At 14 days from intravenous injection of $5 \times 10^5$ B16F10 melanoma cells into WT and Chi3l1 KO mice, metastatic melanoma colonies were observed on the lung surface (Fig. 6a). Melanoma lung metastasis was significantly reduced in the Chi3l1 KO mice compared to WT animals (Fig. 6b). Histological analysis of sectioned slides revealed that infiltrated tumors around blood vessels were also significantly decreased in Chi3l1 KO mice (Fig. 6c). Increased infiltration of IFNγ-producing CD4 T cells and CD8 T cells was significantly higher in the lungs of Chi3l1 KO mice than WT animals (Fig. 6d, e). However no difference was observed in the IFNγ-producing properties of NK cells or non-lymphocytic populations (Fig. 6f, g). Consistent with the RNA-sequencing and RT-PCR results, Chi3l1 KO mice had

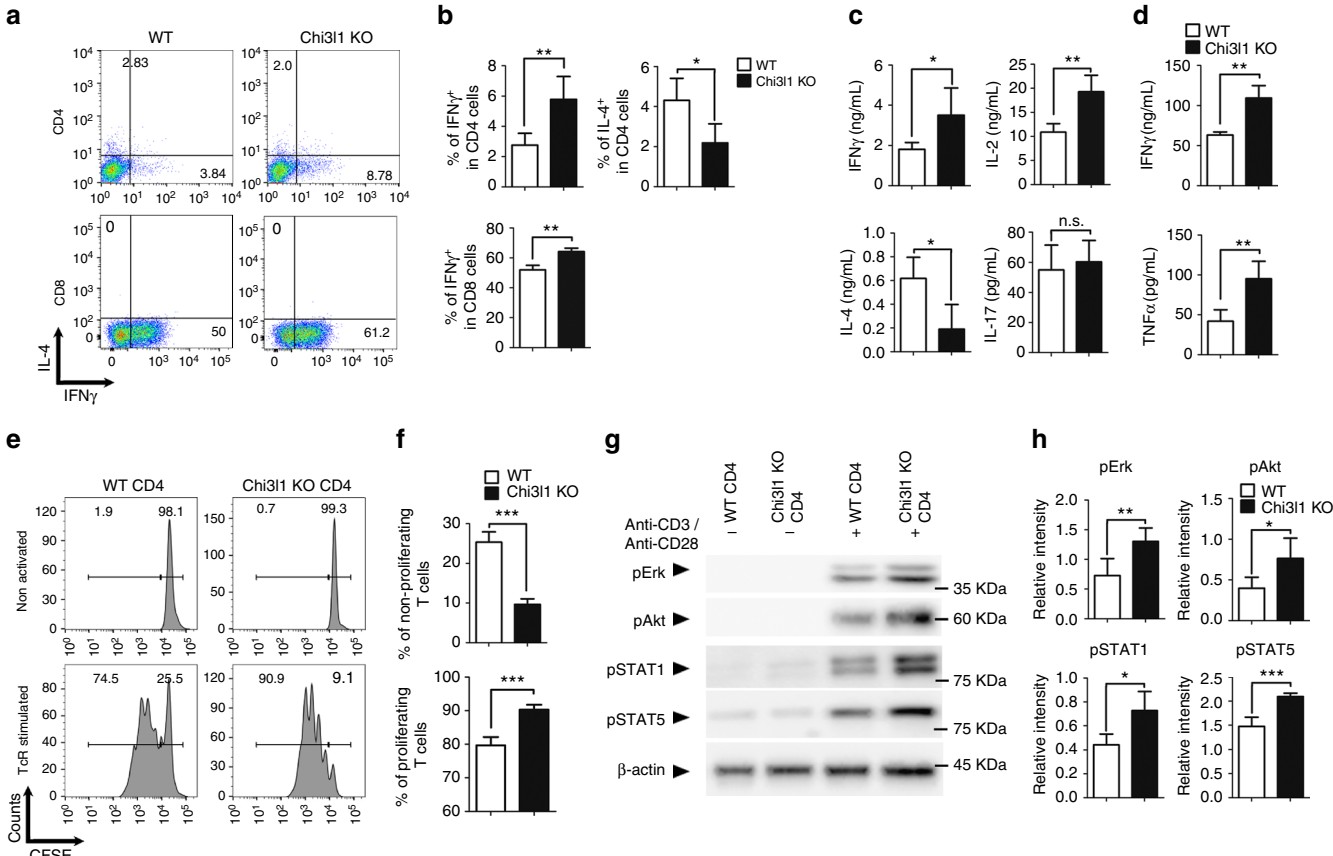

**Fig. 2** Chi3l1 KO T cells are hyper-responsive to TcR stimulation. **a, b** MACS-sorted WT and Chi3l1 KO naïve CD4 T cells were activated by plate-bound anti-CD3 and anti-CD28 antibodies for 3 days. IFNγ and IL-4 expression level was analyzed by flow cytometry. **c** IFNγ, IL-2, IL-4, and IL-17 cytokine production in the CD4 T cell culture supernatant was measured by ELISA. **d** IFNγ and TNFα cytokine production in the CD8 T cell culture supernatant was measured by ELISA. **e** Proportion of proliferating CD4 T cells were analyzed by CFSE assay. **f** Percentages of proliferating and non-proliferating cells were analyzed by gating on histogram. **g** Phosphorylated Erk, Akt, STAT1, STAT5 were analyzed by Western blotting. **h** Relative densitometric analysis of Western blots was represented as normalized to β-actin. Data are mean ± SD of three sets of independent experiments ($n = 6$). n.s., not significant; *$p < 0.05$, **$p < 0.01$, ***$p < 0.001$ (two-tailed Student's $t$-test)

increased mRNA for anti-tumor immunity molecules including CTSE, TRAL, IFNγ, T-bet, Perforin, and Granzyme B, while the expression of Th1-inhibitory molecules such as Twist1 and SOCS1 was significantly reduced (Fig. 6h). To confirm the increased tumoricidal activity in Chi3l1 KO CD8 T cells, we performed co-culturing experiments with pre-activated WT or Chi3l1 KO CD8 T cells and B16F10 melanoma cells. Chi3l1 KO CD8 T cells showed more potent tumor-killing activity than WT CD8 T cells (Fig. 6i). However, no significant difference in the cytotoxicity of WT and Chi3l1 KO NK cells was observed (Fig. 6j). This result suggest reduction of melanoma lung metastasis would be mainly dependent on altered T cell responses. To further confirm the question whether inhibition of pulmonary metastasis is truly depending on altering T cell functions, we generated CD4 specific Chi3l1 KO mice by crossing CD4-Cre and Chi3l1 floxed mice (Fig. 7a). At 14 days from melanoma tumor cell injection, as consistent, we found significantly decreased number of pleural colonies in the lung of CD4-Chi3l1 KO mice (Fig. 7b, c) with reduced tumor infiltration around the vessel (Fig. 7d). There are significantly increased IFNγ and TNFα expressing both CD4 and CD8 T cells (Fig. 7e, f), while there are no differences in NK cells and non-lymphocytic populations in CD4-Chi3l1 KO mice (Fig. 7g, h). Very consistently, there are elevated anti-tumor molecules such as perforin, granzyme B, CTSE and increased Th1 transcription factor T-bet while decrease of Th1 inhibitory molecule like SOCS1 and

Twist1 in the lung of CD4-Chi3l1 KO mice (Fig. 7i). Importantly we further confirmed that there were significantly increased tumor infiltrating lymphocyte (TIL) populations in the lung by CD45 staining which contains IFNγ producing T cells with increased Th1 and CTL related mRNA expression in tumor region. Although there is no significance of statistical analysis due to the limited number of samples, CD8/Treg ratio showed increased pattern in CD4-Chi3l1 KO mice, which is clinically critical value for successful immunotherapy[18,19] (Supplementary Fig. 3). However, IFNγ producing T cells from spleen and inguinal lymph node was not increased (Supplementary Fig. 4), suggesting elevated antigen specific T cell recruitment and actions. Furthermore, we transferred WT or Chi3l1 KO Th1 and activated CD8 T cells into RAG KO mice consistently shows that Chi3l1 KO T cell transfer shows more significantly reduced lung metastasis of melanoma cells compare to WT T cell transfer (Supplementary Fig. 5) suggesting the total KO and CD4-Chi3l1 KO mice phenotype of reduced lung metastasis would be due to altered intrinsic T cell functions. Collectively, these results suggest that Chi3l1 is a negative regulator of Th1 and CTL responses to enhance pulmonary metastasis.

**Targeted silencing of Chi3l1 by peptide-siRNA complex**. Previously, we reported a novel cell-penetrating peptide (CPP), dNP2, which delivers an immune regulatory protein into T cells

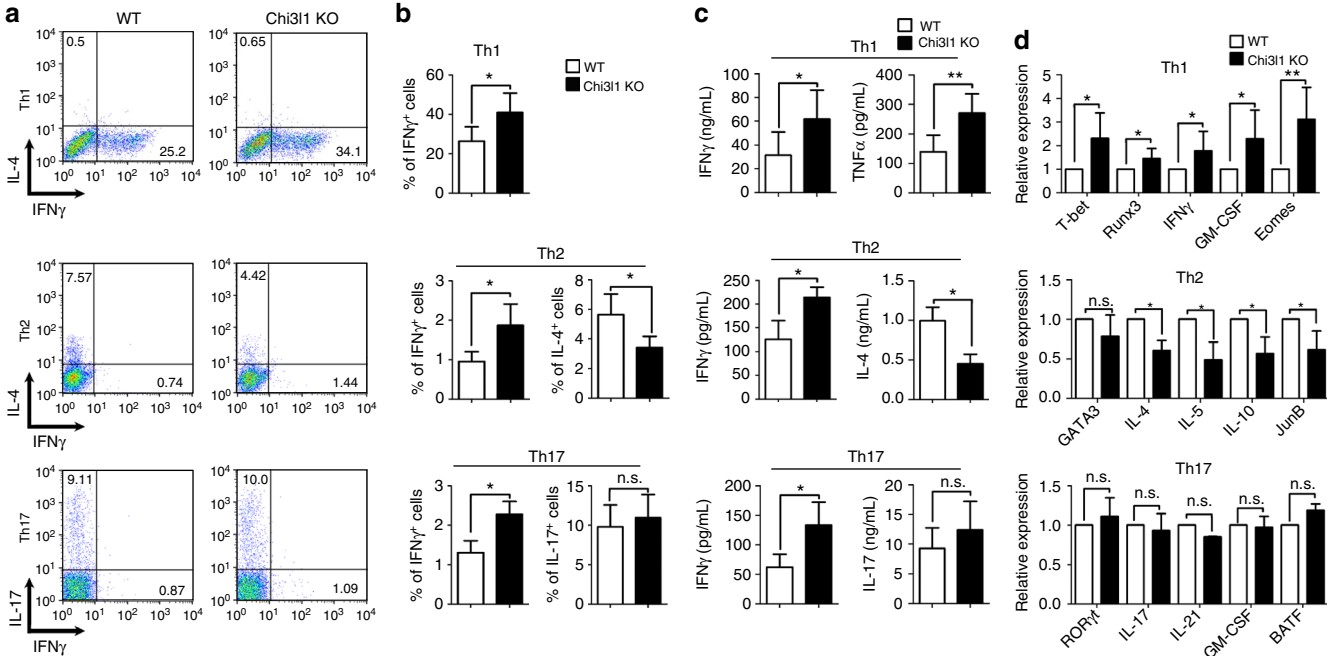

**Fig. 3** Chi3l1 KO CD4 T cells are prone to Th1 differentiation. **a**, **b** MACS-sorted WT and Chi3l1 KO naïve CD4 T cells were differentiated into Th1, Th2, and Th17 cells under specific cytokine-skewing condition, and Intracellular cytokine staining performed to analyze lineage-specific cytokine expression. **c** IFNγ, TNFα, IL-4, and IL-17 production in culture supernatants were measured by ELISA. **d** Lineage specific mRNA levels were analyzed by quantitative RT-PCR. Each gene expression level was normalized to β-actin, and represented as relative expression to WT. Data are mean ± SD of eight sets of independent experiments (n = 16). n.s., not significant; *p < 0.05, **p < 0.01 (two-tailed Student's t-test)

and showed its therapeutic potential in autoimmune diseases[20,21] We hypothesized that introducing siRNA against Chi3l1 (siChi3l1) via dNP2 peptide would be a novel strategy and could enhance anti-tumor immunity to reject tumors. We utilized chemically synthesized dNP2 peptide combined with HA2 for helping endosomal escape[22] for making noncovalent peptide-siRNA complex based on N/P ratio, which refers the number of amine group in dNP2-HA2 to the number of phosphate group in siRNA (Fig. 8a). 30 min incubation at room temperature of the complex was analyzed by gel retardation assay that dNP2-HA2 start to form complexes with siChi3l1 from 1:10, and mainly at 1:25 N/P ratio (Fig. 8b), thus we chose dNP2-siRNA complex with 1:25 N/P ratio for further experiments. The dNP2-siChi3l1 complex size was measured by Nano particle size analyzer which shows the size of the complex is increasing as the N/P ratio (Fig. 8c). The molecular complex of 1:25 indicates around 300 nm. To confirm the knockdown efficiency of the complexes, we used Chi3l1-overexpressed HEK293T cells. The knockdown efficiency of dNP2-siChi3l1 complex or Lipofectamine-mediated siChi3l1 delivery was compared at the mRNA (Fig. 8d) and protein levels (Fig. 8e). dNP2-mediated siChi3l1 delivery significantly suppressed Chi3l1 expression dose dependently and the level of suppression at 250 ng siRNA at 1:25 N/P ratio was comparable to Lipo-siChi3l1 positive control. The siRNA complex of dNP2-siChi3l1 and dNP2-siEGFP was intranasally administered to C57BL/6 mice at day 0, and in vivo Chi3l1 knockdown efficiency was evaluated by RT-PCR. 2.5 μg of siChi3l1 delivery by 70 μg of dNP2-HA2 peptide at 1:25 N/P ratio showed significant reduction of Chi3l1 mRNA expression in the lung at day 1. Maximum silencing of Chi3l1 expression was noted at day 2 and the level of Chi3l1 started to restore from day 3 (Fig. 8f). Chi3l1 protein was significantly reduced from day 2 and was sustained until day 3 (Fig. 8g, h). Both mRNA and protein level of Chi3l1 in the lung was reduced by dNP2-siChi3l1 in dose dependent manner (Supplementary Fig. 6a, b), however there was

no significant effect observed by free siChi3l1 or free dNP2-HA2 peptide (Supplementary Fig. 6c) suggesting dNP2-HA2 peptide truly enables siChi3l1 delivery into lungs via intranasal route and knockdown target gene expression. Next, we determined the functional activity of dNP-siChi3l1 in Th1 differentiation. FACS-sorted WT naïve CD4 T cells were differentiated into Th1 cells with dNP2-siChi3l1 or dNP2-siEGFP for 5 days. IFNγ- and TNFα-producing cells were analyzed by flow cytometry (Fig. 8i, j). From 100 ng to 250 ng, dNP2-siChi3l1 resulted in significantly increased IFNγ-producing and TNFα-producing cells compared to a dNP2-siEGFP-treated group. The increase of IFNγ mRNA level by dNP2-siChi3l1 has correlation with significantly decrease of Chi3l1 mRNA level in Th1 cells (Fig. 8k). We confirmed there was no effect of equivalent dose of free siChi3l1 treatment in Th1 differentiation (Supplementary Fig. 7) suggesting critical role of dNP2-HA2 for delivery of siRNA into T cells. These studies collectively elicit that siRNA targeting Chi3l1 could be delivered into T cells by dNP2-HA2 peptide and strengthened Th1 differentiation and enhanced effector cytokine production by activated CD8 T cells which would be a novel strategy to modulate T cell immunity.

**dNP2-siChi3l1 treatment suppresses pulmonary metastasis.** Finally, we investigated the in vivo efficiency of dNP2-siChi3l1 treatment in the regulation of melanoma lung metastasis. 14 days after intravenous injection of 5 × 10⁵ B16F10 melanoma cells into WT mice, lung tissues were harvested and analyzed. 2.5 μg control siEGFP and siChi3l1 complexed with 70 μg dNP2-HA peptide at 1:25 N/P ratio were intranasally administered twice per day for every other day from day 0 to day 14 (Fig. 9a). Metastatic melanoma colonies on the lung surface were observed (Fig. 9b) and counted. There was a significant reduction of number of metastatic pleural colonies (Fig. 9c) and relative size of melanoma colony (Fig. 9d) in the lung by dNP2-siChi3l1 treatment

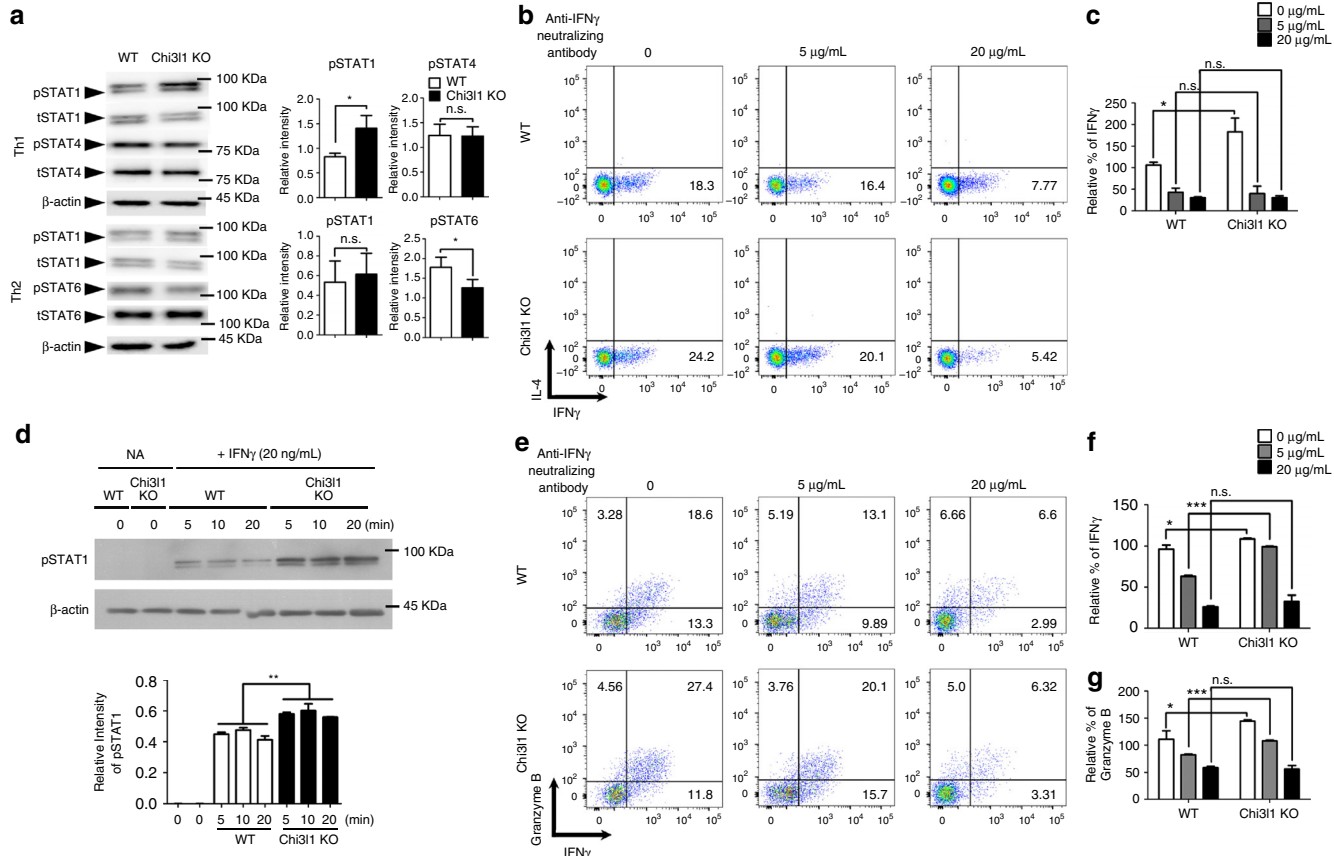

**Fig. 4** Increased Th1 and CTL response in Chi3l1 KO T cells are IFNγ dependent. **a** Phosphorylation of STAT1, STAT4, STAT6 were analyzed by Western blotting. Relative densitometric analysis of Western blotting normalized to total STAT. **b** WT and Chi3l1 KO naive CD4 T cells were differentiated into Th1 cells with the indicated concentrations of IFNγ neutralizing antibodies and assessed for IFNγ and IL-4 expression by intracellular cytokine staining. **c** Proportion of IFNγ producing cells were represented by relative % of IFNγ to WT Th1. **d** FACS-sorted WT and Chi3l1 KO naïve CD4 T cells were treated with IFNγ and pSTAT1 level was analyzed by Western blotting. Densitometric values of band intensity was calculated by normalization to the value of β-actin. **e** Intracellular level of IFNγ and Granzyme B in CD8 T cells. **f** Proportion of IFNγ producing cells were represented as relative % of IFNγ to WT CD8 T cells without anti-IFNγ neutralizing antibody. **g** Proportion of Granzyme B expressing cells were represented as relative % of Granzyme B to WT CD8 T cells without anti-IFNγ neutralizing antibody. Data are mean ± SD of three sets of independent experiments ($n = 6$). n.s., not significant; *$p < 0.05$, ***$p < 0.001$

compared to dNP2-siEGFP-treated group. Histological analysis of sectioned slides revealed that infiltrated tumors around blood vessels were significantly decreased in dNP2-siChi3l1 mice (Fig. 9e). In addition, increased infiltrated IFNγ-producing CD4 T cells and Granzyme B-producing CD8 T cells were noted in the lungs by dNP2-siChi3l1 (Fig. 9f, g). The protein level of intracellular T-bet in both IFNγ-producing CD4 and CD8 T cells was significantly higher by dNP2-siChi3l1 treatment than the mice with dNP2-siEGFP treatment (Fig. 9h). However, no difference was noted in NK1.1 positive NK cells and NK1.1−CD4−CD8− cells on IFNγ or Granzyme B expression by dNP2-siChi3l1 treatment (Supplementary Fig. 8) suggesting peptide-siRNA complex treatment specifically influences T cell functions. Consistent with the RNA-sequencing data and RT-PCR results, dNP2-siChi3l1-treated mice showed increased gene expression of anti-tumor immunity molecules CTSE, TRAIL, IFNγ, T-bet, Perforin, and Granzyme B, while the levels of Th1-inhibitory molecules such as Twist1 and SOCS1 were significantly down-regulated (Fig. 9i). Free siChi3l1 treatment without dNP2-HA2 peptide does not effect on melanoma lung metastasis and cytokine productions by T cells (Supplementary Fig. 9) suggesting dNP2-HA2 complex is critical for inhibition of metastasis. Furthermore, when we started to treat dNP2-siChi3l1 intranasally at 2 days after intravenous tumor injection as a therapeutic scheme, it also still significantly

decreased the number of tumor colonies in the lung with increased expression of IFNγ and TNFα expressing CD4 and CD8 T cells suggesting it could inhibit tumor growth in the lung as a therapeutic agent (Supplementary Fig. 10). Collectively, these data suggest that dNP2-siChi3l1 is an effective and a novel siRNA complex agent significantly enhances anti-tumor immunity of Th1 and CTL responses to inhibit tumor metastasis or growth in the lung.

## Discussion

Chitinase-like protein is an evolutionarily conserved protein in mammals and does not show enzymatic activity[4,5], while chitinase is an enzymatic defense system against pathogens in lower organisms[23,24]. In mammals, Chi3l1, BRP-39, and YKL-40 are expressed by cells including macrophages, chondrocytes, and synovial cells and are involved in inflammation and tissue remodeling[6,25]. Biological and physiological studies have shown that macrophage expression of Ym-1 is highly induced by IL-4 and STAT6, which are recognized as M2 macrophage markers, and induced Ym-1 promotes Th2 cytokine production and airway inflammation[7,8,17,26]. In addition, Ym-1/2 induce γδ T-cell-mediated neutrophil migration and IL-17 production against nematode infection in mice[9]. Since no human homologs of Ym-1/

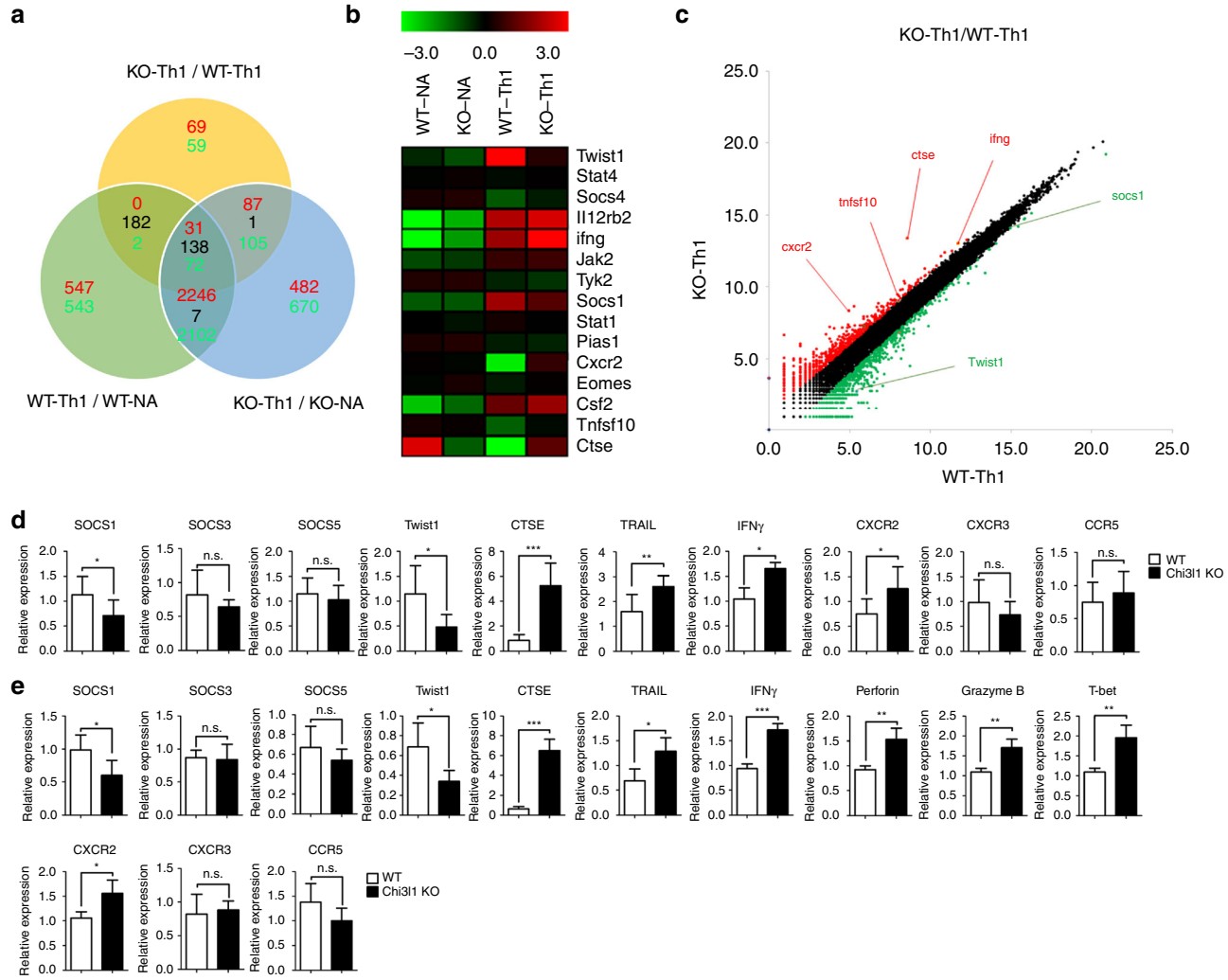

**Fig. 5** RNA transcriptomes of Chi3l1 KO T cells. **a** 100-bp pair-ended RNA-sequencing in naïve, Th1-skewed WT, and Chi3l1 KO CD4 T cells. The number of up-regulated genes were indicated as red, and down-regulated genes were indicated as green in each comparative analysis. **b** Heatmap analysis of genes of interest. **c** Scatterplot indicates either of over than two-fold up-regulated and down-regulated genes in WT Th1 versus Chi3l1 KO Th1 comparison. **d** Expression of genes of interest related to cytotoxicity and the IFNγ signaling pathway in Th1 cells were confirmed by quantitative RT-PCR. Each gene expression level was normalized to β-actin, and represented as relative expression to WT. **e** Quantitative RT-PCR was performed in activated CD8 T cells. Each gene expression level was normalized to β-actin, and represented as relative to WT. Data are mean ± SD of ten sets of independent experiments ($n = 20$). n.s., not significant; $*p < 0.05$, $**p < 0.01$, $***p < 0.001$ (two-tailed Student's $t$-test)

2 are identified, the application of these Ym-1/2 findings in actual human diseases could be significantly limited. On the other hand, Chi3l1, highly homologous to Ym-1/2, is expressed both in human and rodent species. Importantly, the dysregulated expression of Chi3l1 is also significantly associated with many of human diseases including asthma and various tumors. In recent studies using transgenic or KO mice of Chi3l1 in various animal models of lung diseases showed an essential role of Chi3l1 in the pathogenesis of inflammation and tissue remodeling[6,10,11,27,28]. Chi3l1 expression is highly increased in Th2 inflammatory conditions induced by OVA/Alum and house dust mite (HDM), and IL-13 transgenic mice[6]. Chi3l1 KO mice show reduced Th2 inflammation, and overexpression of YKL-40 reverses this phenotype[6,29]. In addition, IL-13 production by T cells and Th2 inflammation in lung-specific overexpression of IL-18[30] are significantly reduced by Chi3l1 KO, however, IFNγ production by T cells and Perforin, Granzyme B levels are significantly increased by Chi3l1 deficiency[14]. Although these studies identified Chi3l1 as a significant immune modulator, the immune regulatory function of Chi3l1 especially in T cells has not been characterized.

Here, we demonstrate that Chi3l1 is expressed in activated T cells and Th2 cells, then regulates Th1 and Th2 differentiation through increased IFNγ signaling. The genetic and chemical ablation of Chi3l1 in T cells significantly enhances Th1 and CTL responses and inhibits tumor growth and lung metastasis. Our studies also suggest that Chi3l1 regulation of T cell response is a critical event in the pathogenesis of inflammatory and tissue remodeling and tumor growth and progression in which the expression of Chi3l1 is dysregulated.

Previous studies revealed that IL-13Rα2 mediates the Chi3l1 signaling and various effector function of Chi3l1 as a putative receptor[31]. Exogenous Chi3l1 physically interacts with IL-13Rα2, and recombinant Chi3l1 treatment enhances Erk/Akt phosphorylation and β-catenin nuclear translocation in THP1 cells. TMEM219 is reported to be another component of the Chi3l1-IL-13Rα2 complex that physically binds to IL-13Rα2[32]. TMEM219 deficiency dampens the function of the Chi3l1-IL-13Rα2 axis. However, we did not observe any effect of recombinant mouse Chi3l1 (rmChi3l1) treatment on Th1 or Th2 differentiation in vitro, while treatment of rmChi3l1 significantly

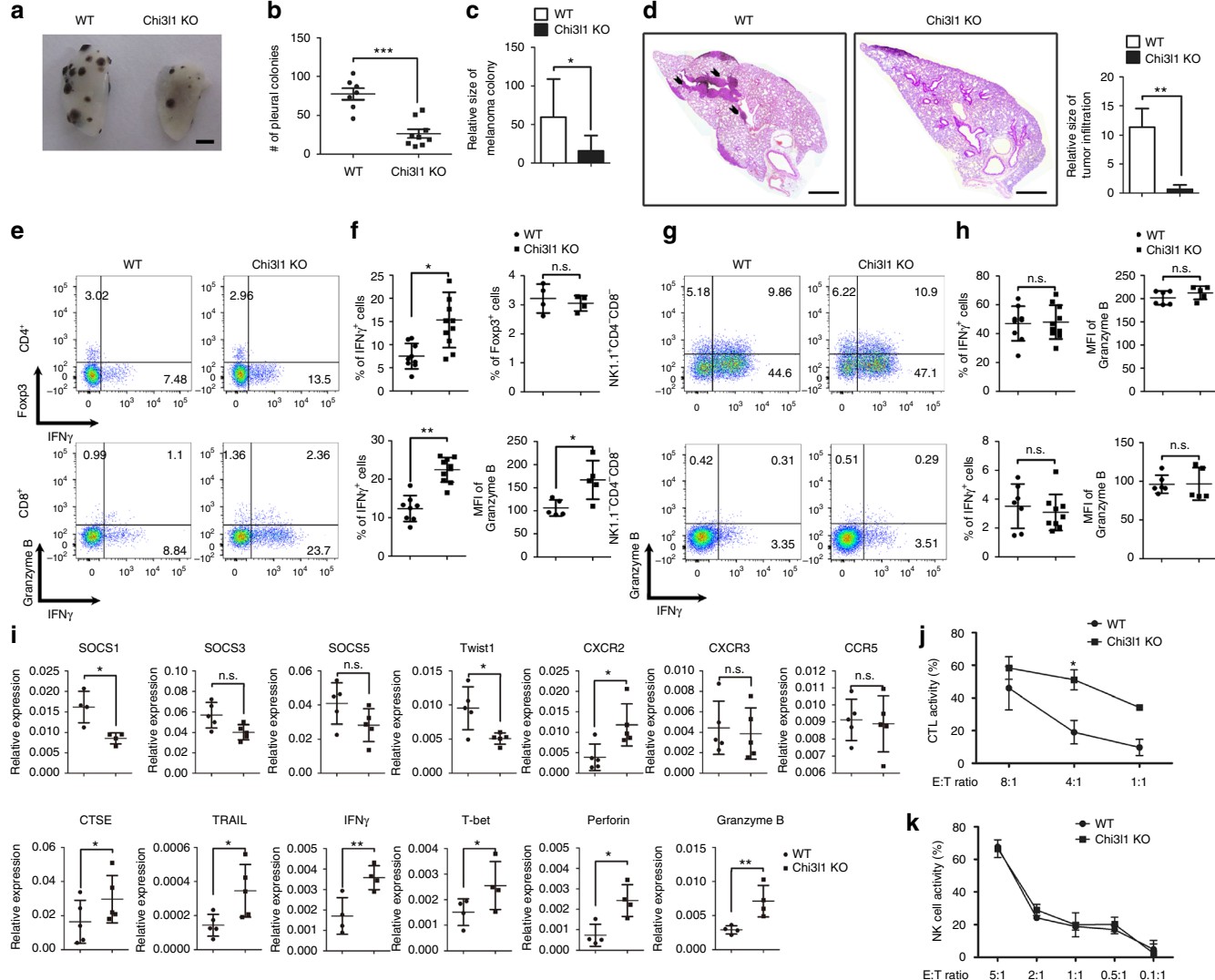

**Fig. 6** Chi3l1 KO mice have reduced pulmonary metastasis with increased IFNγ-producing CD4 and CD8 T cells in the lung. **a** Representative lung image from B16F10 melanoma injected WT and Chi3l1 KO mice. Scale bar, 2 mm **b** Number of pleural colonies in each lung was counted. Data are mean ± SEM of three sets of independent experiments and each dot in graphs represent an individual mouse. **c** Relative total tumor area in the lung was measured by Image J software 1.48 v. **d** Histology of lung sections by H&E staining, and infiltrated tumor region was measured by Image J software 1.48 v. Scale bar, 200 m. **e, f** IFNγ producing CD4 T cells, Foxp3 + regulatory T cells, IFNγ producing CD8 T cells, and Granzyme B expression in IFNγ⁺ CD8 T cells in the lung was analyzed by intracellular cytokine staining. % of IFNγ⁺, % of Foxp3⁺, and MFI of Granzyme B was represented as scattered graph. **g, h** IFNγ producing NK cells IFNγ producing non-lymphocytic population, and Granzyme B expression in IFNγ⁺ NK cells and non-lymphocytic population in the lung was analyzed. % of IFNγ⁺, and MFI of Granzyme B was represented as scattered graph. **i** mRNA expression of genes related to cytotoxicity and Th1 effector functions was analyzed by quantitative RT-PCR. Each gene expression level was normalized to β-actin. **j** Cytotoxicity of WT and Chi3l1 KO CD8 T cells against B16F10 target cells at indicated E:T ratios. **k** NK cell activity was represented as tumor killing activity of WT and Chi3l1 KO NK cells against to B16F10 target cells at indicated E:T ratios. Data are mean ± SD of three sets of independent experiments and each dot in graphs represent an individual mouse. n.s., not significant; *$p < 0.05$, **$p < 0.01$, ***$p < 0.001$ (two-tailed Student's $t$-test)

reduces pro-inflammatory cytokine production in LPS-stimulated peritoneal macrophages (Supplementary Fig. 11). In addition, IL-13Rα2 mRNA was not detected in T cells with a significant phenotype onto dNP2-siChi3l1 treatment, suggesting an alternative biological mechanism of Chi3l1 in regulation of T cell differentiation and activity. Recently Chi3l1 was reported as localized in the cytosol and the nucleus, which could promote monocyte-derived DC maturation[33]. We also speculate a possibility that intracellular Chi3l1 could be able to regulate T cell functions through more direct interaction with the molecules working at the levels of cytosol or nucleus. Based on RNA-sequencing analysis in our study, Twist1 is proposed as a target molecule regulated by Chi3l1. Twist1 is a member of the basic

helix-loop-helix family and has been studied as a negative regulator of Th1 differentiation by suppression of IFNγ production and T-bet and Runx3 functions[34,35]. In addition, Twist1 induces SOCS1, which inhibits STAT1 phosphorylation to regulate the IFNγ signaling pathway[34]. Thus, it is reasonable to speculate that the decreased expression of Twist1 in the Chi3l1-deficient Th1 cells could be responsible for the increased IFNγ signaling and related effector functions of T cells. In turn, increased IFNγ signaling in Chi3l1 KO T cells shows more potent tumor-killing activity with increased tumoricidal expression of molecules such as CTSE, TRAIL, and Granzyme B. Previously, IFNγ has been shown to increase the expression of ctse[36] and trail[37], which contribute to tumor clearance in mice[37–39]. In addition, IFNγ also

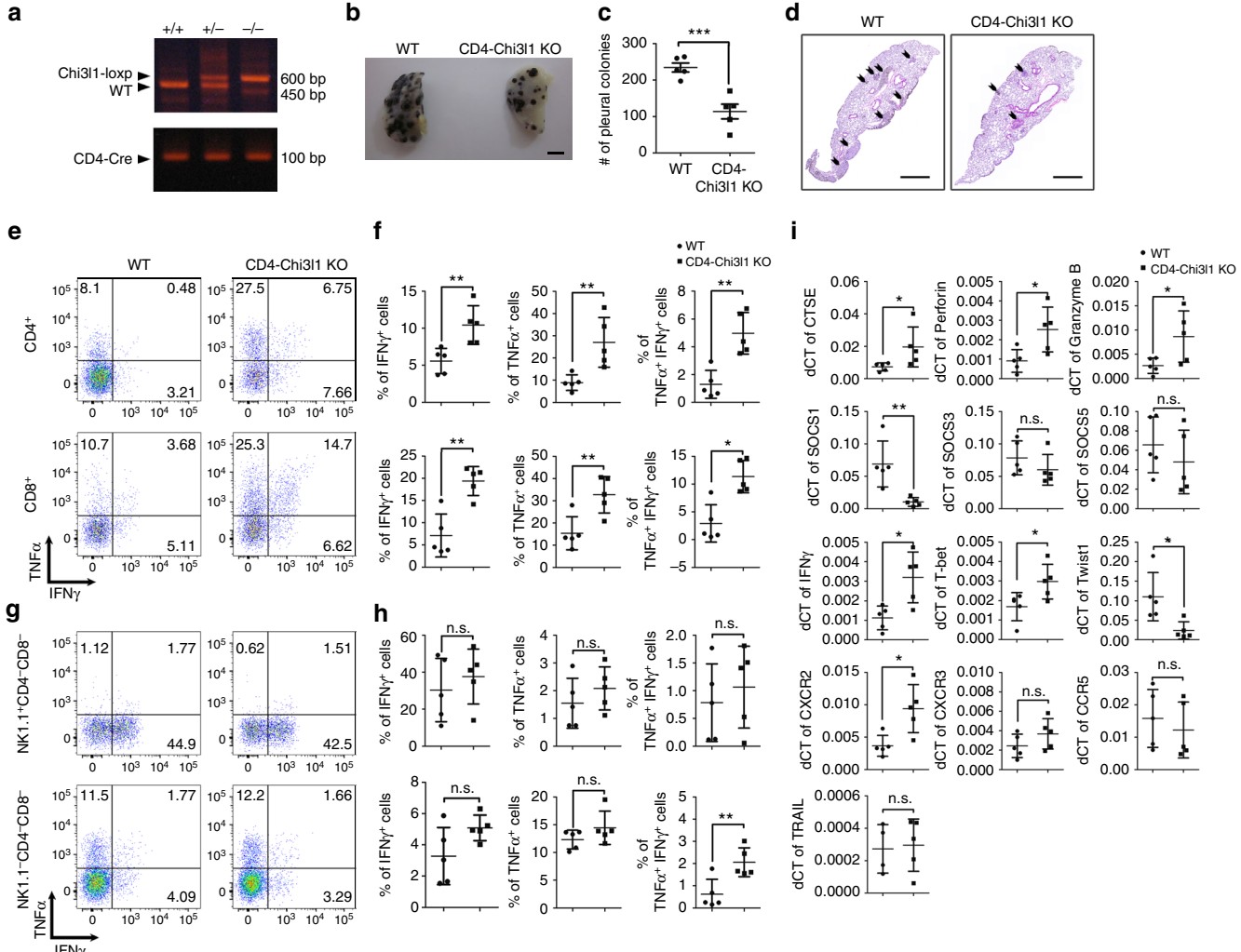

**Fig. 7** Chi3l1 deletion in T cell inhibits pulmonary metastasis with increase of Th1 and CTL. **a** Genotyping PCR to confirm CD4 specific deletion of Chi3l1 in the mice. **b** Representative lung image from B16F10 melanoma injected WT and CD4-Chi3l1 KO mice. Scale bar, 2 mm **c** Number of pleural colonies in each lung was counted. Data are mean ± SEM of three sets of independent experiments and each dot in graphs represent an individual mouse. **d** Histology of lung sections by H&E staining. Scale bar, 200 m (**e**, **f**) IFNγ and/or TNFα producing CD4 T cells and CD8 T cells in the lung was analyzed by intracellular cytokine staining. % of IFNγ⁺, % of TNFα⁺, and % of TNFα⁺IFNγ⁺ was represented as scattered graph. **g**, **h** IFNγ and/or TNFα producing NK cells, and non-lymphocytic populations (NK1.1⁻CD4⁻CD8⁻) in the lung was analyzed by intracellular cytokine staining. % of IFNγ⁺, % of TNFα⁺, and % of TNFα⁺IFNγ⁺ was represented as scattered graph. **i** mRNA expression of genes related to cytotoxicity and Th1 effector functions was analyzed by quantitative RT-PCR. Each gene expression level was normalized to β-actin. Data are mean ± SD of three sets of independent experiments and each dot in graphs represent an individual mouse. n.s., not significant; *$p < 0.05$, **$p < 0.01$, ***$p < 0.001$ (two-tailed Student's t-test)

was reported to induce the expression of T-bet, which promotes Perforin and Granzyme B expression in CD8 T cells[40,41]. However, we did not observe significant functional differences in Chi3l1 KO NK cells compared to WT NK cells in tumor-killing activity or effector molecule expression in vitro and in vivo. As presumably mRNA expression of Twist1 was significantly lower in NK cells than T cells, no difference was noted in the expression of Twist1, T-bet, IFNγ, Granzyme B, and Perforin between WT and Chi3l1 KO NK cells (Supplementary Fig. 12). These studies further suggest that T cells are potentially major drivers of IFNγ and other cytotoxic gene expression associated with anti-tumor activity in the absence of Chi3l1. The exact molecular mechanism of Chi3l1 in the expression of Twist1 and other genes need to be further defined in the future studies.

Chi3l1 was discovered in mouse breast cancer cells and has been studied regarding tumors and inflammatory diseases[6,10,11,27,28]. Chi3l1 is elevated in patients with a variety of

tumors, and recent studies have suggested that Chi3l1 promotes cancers by enhancing cell proliferation, angiogenesis, etc[28,42–44]. In addition, recent studies have shown that Chi3l1 that is induced by the Semaphorin7A-β1 integrin signaling pathway contributes to establishment of a metastatic microenvironment in the lung[10], and RIG-like Helicase activation reduces Chi3l1 expression, which results in suppression of tumor progression in the lung[11]. More recently, it was reported that Chi3l1 by fibroblast enhances migration and growth of breast cancer, and knockdown of Chi3l1 by shRNA increases T cell population and IFNγ, TNFα expression[45]. Here, we added a novel regulatory function for intrinsic Chi3l1 in anti-tumor T cell immunity in pulmonary metastasis through functional regulation of Th1 cells and CTL activity. Thus, these findings further suggest that Chi3l1 is an important therapeutic target for treating tumors with dysregulated expression of Chi3l1. Recently, IL-4 and Th2 inflammation was shown to promote B16F10 melanoma metastasis[15]. Since Chi3l1 deficiency

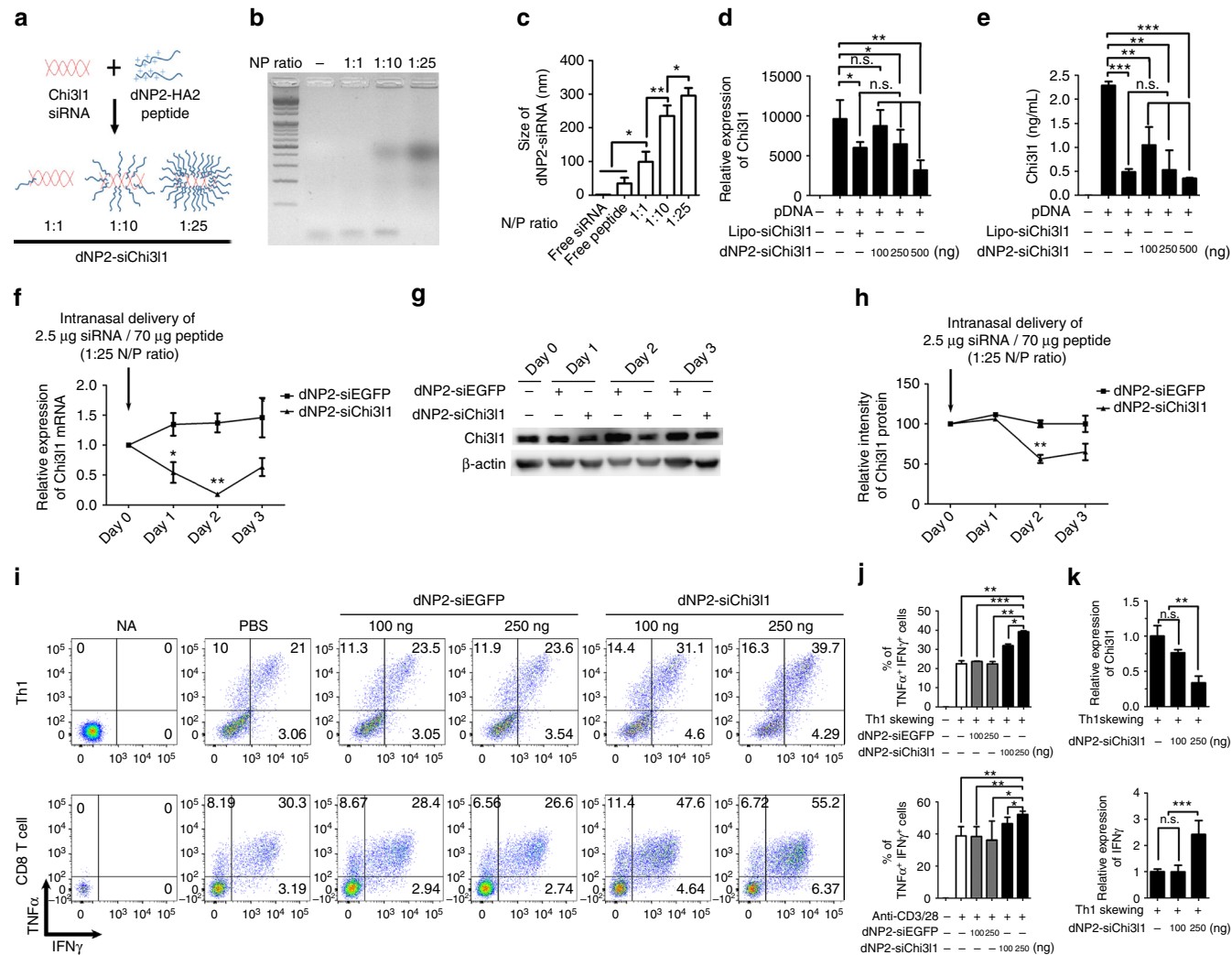

**Fig. 8** Targeted inhibition of Chi3l1 by dNP2-siChi3l1 complex enhances effector cytokine production in Th1 and CD8 T cells. **a** Schematic diagram of the complex formation of dNP2-HA2 peptide and Chi3l1 siRNA (siChi3l1) at indicated N/P ratios. **b** Gel retardation assay of dNP2-siChi3l1 complex. **c** The size of free siRNA, free peptide, and dNP2-siRNA complexes were measured by Nano particle size analyzer. **d** Dose dependent reduction of Chi3l1 mRNA expression by dNP2-siChi3l1 complex at 1:25 N/P ratio. **e** Chi3l1 protein level in culture supernatants were measured by ELISA. **f** Chi3l1 mRNA in the lung after intranasal administration of dNP2-siEGFP or dNP2-siChi3l1 complex at 1:25 N/P ratio. Chi3l1 mRNA normalized to β-actin, and presented as relative to expression at day 0. **g** Chi3l1 protein level in the lung was analyzed by Western blotting. Chi3l1 protein relative to expression at day 0. **h** Densitometric values of band intensity was calculated by normalization to the value of β-actin, then the value was represented as relative to the value at day 0. Statistical significance of dNP2-siChi3l1 treated group was analyzed to dNP2-siEGFP treated group on each day. **i** FACS-sorted WT naïve CD4 T cells were differentiated into Th1 cells and WT naïve CD8 T cells were activated by plate-bound anti-CD3 and anti-CD28 antibodies with IL-2 for 5 days with indicated concentrations of dNP2-siEGFP or dNP2-siChi3l1 complexes. IFNγ and TNFα expression were analyzed by flow cytometry. **j** % of TNFα⁺IFNγ⁺ cells were represented as bar graph. **k** Chi3l1 and IFNγ mRNA expression in dNP2-siChi3l1 treated Th1 cells. Each gene expression level was normalized to β-actin, and represented as fold change to those of WT Th1 cell. Data are presented as mean ± SD of three sets of independent experiments ($n = 6$). n.s., not significant; $*p < 0.05$, $**p < 0.01$, $***p < 0.001$ (two-tailed Student's $t$-test)

reduced IL-4 production in Th2 cells while increasing IFNγ expression in Th1 and CD8 T cells, targeting Chi3l1 could provide prominent anti-tumor effect by driving Th1 response while suppressing the activity of tumor-promoting Th2 cells. In addition, dNP2-siChi3l1 treatment did not inhibit proliferation of melanoma cells (Supplementary Fig. 13), suggesting that its in vivo inhibitory effects on lung metastasis were mainly due to the enhanced anti-tumor immunity, not by inducing direct tumor cell death or suppressing proliferation of tumor cells.

Recently, cancer immunotherapy has been highlighted and clinically validated for many cancers which enhances tumor-killing T cell functions including immune checkpoint blockade such as anti-PD-1 and anti-CTLA-4 antibodies, engineered T cell therapy utilizing chimeric antigen receptors (CAR), and cancer

vaccines, etc[46–50]. Although cancer immunotherapy has been emerging as an important addition to conventional therapies, still there have been reported side effects or limitations of current approaches to have autoimmune responses in significant populations of patients or less effective due to individual variabilities of tumor antigens[47,49,51]. Although we do not know whether Chi3l1 is implicated in the regulation of immune checkpoint molecules such as PD-1 in the T cells, it is intriguing to speculate that Chi3l1 could complement the immunotherapy against the tumors that do not respond to current immunotherapy or other approaches. Based on Chi3l1 KO mouse phenotypes in the B16F10 melanoma lung metastasis model, we developed a novel tumor treatment agent based on siRNA and cell-penetrating peptide (CPP) complexes. Previously, we identified and reported novel human-

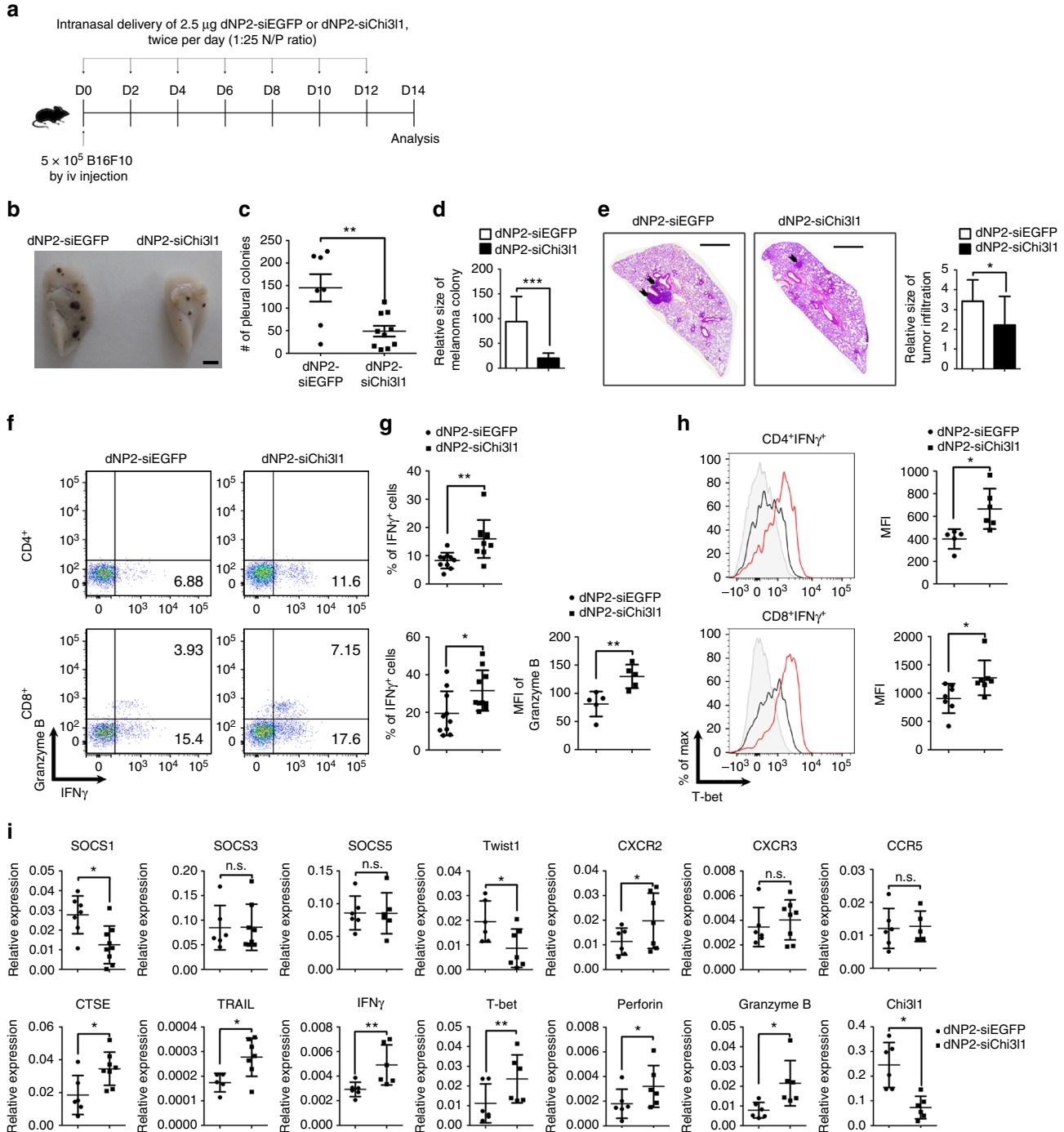

**Fig. 9** In vivo administration of dNP2-siChi3l1 complex inhibits pulmonary metastasis with enhanced Th1 and CTL effector molecules. **a** Experimental scheme of dNP2-siRNA complex treatment in pulmonary melanoma metastasis model. **b** Representative lung image of dNP2-siEGFP and dNP2-siChi3l1-treated mice. Scale bar, 2 mm **c** Number of pleural colonies in each lung was counted. Data are mean ± SEM of three sets of independent experiments and each dot in graphs represent an individual mouse. **d** Relative total tumor area in the lung was measured by Image J software 1.48 v. **e** Histology of lung sections by H&E staining, and infiltrated tumor region was measured by Image J software 1.48 v. Scale bar, 200 m **f**, **g** IFNγ producing CD4 and CD8 T cells, and Granzyme B expression level in IFNγ⁺ CD8 T cells in the lung was analyzed by intracellular staining. % of IFNγ⁺, and MFI of Granzyme B was represented as scattered graph. **h** Intracellular T-bet expression level in CD4⁺IFNγ⁺ and CD8⁺IFNγ⁺ population. MFI was represented as scattered graph. **i** mRNA expression of genes related to cytotoxicity and Th1 effector functions was analyzed by quantitative RT-PCR. Each gene expression level was normalized to β-actin. Data are mean ± SD of three sets of independent experiments and each dot in graphs represent an individual mouse. n.s., not significant; *p < 0.05, **p < 0.01, ***p < 0.001 (two-tailed Student's t-test)

derived CPPs such as LPIN3[52], 2pIL-1αNLS[53] and their application potentials[54]. Recently, we identified another human-derived CPP, dNP2 that efficiently escapes blood vessels and localizes into resident cells in tissues differently from other CPPs

such as TAT or Hph-1[20]. In addition, we showed it could be efficiently delivered into lymphocytes including T, B, and NK cells which are hardly transfected cells[21]. Here we utilized dNP2 peptide for effective siRNA delivery targeting Chi3l1 in tumor

lung metastasis. We demonstrated that intranasal delivery of the siRNA with dNP2-CPP significantly enhanced in vivo Th1 and CTL activity with increased effector molecules in the lung but not in other peripheral tissues (Supplementary Fig. 4), that efficiently reduced lung metastasis of B16F10 melanoma cells. We also preliminary attempted subcutaneous injection of B16F10 cells that we still found significantly increased tumor-infiltrating IFNγ and/or TNFα producing CD4 and CD8 T cells although tumor size was not reduced (Supplementary Fig. 14). As a further study, we will more investigate antigen specific solid tumor model such as EG-7-OVA versus OT-II TcR transgenic background CD4-Chi3l1 KO mice. We also speculate there would be synergistic function of increased Th1 immunity with macrophages in vivo because dNP2-siChi3l1 could enhance IL-12 or IL-6 production in LPS-stimulated macrophages in vitro (Supplementary Fig. 15). Collectively, our observation suggests that siRNA complex methodology with dNP2-HA2 peptide could be a novel approach and an efficient way of siRNA delivery to T cells and intervene Chi3l1 expression for controlling tumor growth and progression.

In summary, we firstly revealed the function of an evolutionally conserved chitinase-like protein, Chi3l1, in T cell adaptive immunity. We suggest that Chi3l1 negatively regulates Th1 and CTL functions and dNP2-siChi3l1 complex could be an alternative cancer immunotherapy approach for lung metastasis to enhance Th1 and CTL functions but decrease of Th2. The application and potency against various cancers will be further investigated with combination of PD-1 antibody.

## Methods

**Mice**. All mice used in this study were 6-week-old to 8-weeks-old mice on C57BL/6 J background. Chi3l1 KO, and Chi3l1 floxed mice were originally generated and kindly provided by Jack A. Elias. All mice were housed and maintained in a specific pathogen-free facility. All mice were bred in controlled conditions of temperature ($21 \pm 1$ °C), humidity ($50 \pm 5\%$), and 12 h light/dark cycle with regular chow (PicoLab Rodent Diet) and autoclaved water. Two-to-twelve mice that were used in individual experiments were assigned randomly to the experimental groups. Also, both sex of mice was also randomly assigned to the experimental groups. Experiment and data analysis was performed without exclusion criteria. Blinding was impossible in most of animal experiments. All animal protocols used in this study were approved by the Animal Experimentation Ethics Committee. Experiments were performed according to the guidelines of the Institutional Animal Care and Use Committees.

**In vitro T cell activation and differentiation**. WT and Chi3l1 KO naïve CD4 T cells were isolated using mouse CD4⁺CD25⁻CD62L⁺ T Cell Isolation Kit II (Miltenyi Biotec, 130-093-227, 130-104-453) according to the manufacturer's protocols. Purity was around 95%. Fluorochrome-labeled CD4⁺CD25⁻CD62LʰⁱᵍʰCD44ˡᵒʷ naïve CD4 T cells and CD8⁺CD25⁻CD62LʰⁱᵍʰCD44ˡᵒʷ naïve CD8 T cells were sorted by FACS Aria cell sorter (BD Biosciences, Franklin Lakes, NJ). Purity was around 98%. Purified naïve CD4 and CD8 T cells were activated by 2 μg/mL plate-bound anti-CD3 (BD, 145-2C11) and anti-CD28 (BD, 37.51) antibodies and differentiated by exposure to the following cytokines for 3 or 5 days: media only for Th0; IL-2 (50 U/mL, Peprotech) for CTL; IL-12 (0.2 ng/mL, BD Biosciences), IL-2 (50 U/mL), and anti-IL-4 antibody (5 μg/mL, BD Biosciences, 11B11) for Th1; IL-4 (30 ng/mL, BD Biosciences), IL-2 (50 U/mL), and anti-IFNγ antibody (5 μg/mL, BD Biosciences, XMG1.2) for Th2; and TGF-β (0.5 ng/mL, R&D Systems), IL-6 (30 ng/mL, BD Biosciences), IL-23 (20 ng/mL, BD Biosciences), IL-1β (20 ng/mL, R&D Systems), anti-IFNγ (5 μg/mL), and anti-IL-4 (5 μg/mL) antibodies for Th17. For measuring IL-4 expression in the Th2 subset, Th2 cells were harvested at day 5 and further reactivated by plate-bound anti-CD3 antibody for 24 h. In some experiments, the indicated concentration of rmChi3l1 protein or IFNγ-neutralizing antibody or the indicated amount of dNP2-siEGFP or dNP2-siChi3l1 was added to Th1- or CTL-skewing conditions. IL-2, IFNγ, IL-4, and IL-17 were measured in culture supernatants by ELISA (eBioscience).

**In vitro macrophage activation**. Peritoneal macrophages from WT and Chi3l1 KO mice were obtained by the peritoneal lavage. Macrophages stimulated by 10 ng/mL of LPS (O55:B5, sigma L2880) for 12 h with 250 ng of dNP2-siChi3l1 or 100 ng of rmChi3l1 treatment. After LPS stimulation, culture supernatant was analyzed by IL-6 and IL-12 ELISA following manufacturers instruction.

**Flow cytometry**. To measure cell proliferation, sorted WT and Chi3l1 KO naïve CD4 T cells were stained with CFSE (0.75 μM, Invitrogen) for 10 min at 37 °C and washed with complete RPMI media. CFSE-labeled cells were stimulated with plate-bound anti-CD3 and anti-CD28 antibodies for 3 days. Dividing cells were analyzed by flow cytometry, following staining with anti-mouse CD4⁻PerCP-Cy5.5 antibodies (1:1000 diluted, eBioscience, 45-0042-82). To determine intracellular cytokine levels, cells were restimulated with eBioscience Cell Stimulation Cocktail (plus protein transport inhibitors) for 4 h and stained with anti-mouse CD4-PerCP-Cy5.5 or CD8⁻PerCP-Cy5.5 (1:1000 diluted, eBioscience, 45-0081-82) for 15 min. Cells were further fixed, permeabilized, and stained with anti-mouse IFNγ-FITC (1:500 diluted, eBioscience, 11-7311-82), IL-4-PE (1:200, Biolegend, 504103) or TNFα-PE (1:500 diluted, eBioscience, 12-7321-82), and IL-17-APC (1:500 diluted, eBioscience, 17-7177-81) for CD4 T cells or IFNγ-APC (1:500 diluted, eBioscience, 17-7311-81), TNFα-PE (1:500 diluted), Granzyme B-FITC (1:200 diluted, eBioscience, 11-8898-80) for CD8 T cells. Cells were analyzed by flow cytometry. For the pulmonary melanoma metastasis model, purified lymphocytes were restimulated with eBioscience Cell Stimulation Cocktail (plus protein transport inhibitors) for 4 h and stained with anti-mouse CD4-APC-Cy7 (1:200 diluted, BD, 561830), CD8-PerCP-Cy5.5 (1:500, diluted), and NK1.1-PE (1:500 diluted, Biolegend, 108707) or NK1.1-PE-Cy7 (1:500 diluted, Biolegend, PK136), or anti-mouse CD4-APC-Cy7, CD8-PerCP-Cy5.5, and CD45-PE-cy7 (1:500 diluted, Biolegend, 30-F11) for 30 min. Cells were fixed, permeabilized, and stained with anti-mouse IFNγ-APC (1:500 diluted), Granzyme B-FITC (1:200 diluted), and T-bet-PE-cy7 (1:100 diluted, eBioscience, 25-5825-80), Foxp3-PE (1:400 diluted, eBioscience, 12-5773-80) or anti-mouse IFNγ-APC, Granzyme B-FITC, and TNFα-PE (1:500 diluted) and analyzed by flow cytometry. Data analysis was performed without exclusion criteria.

**Western blotting**. Enriched naïve CD4 T cells were skewed to Th0, Th1, and Th2 helper T cells for 3 or 5 days and lysed with RIPA buffer (Cell Signaling Technology) in the presence of HALT inhibitor (Thermo Fisher Scientific) on ice for 30 min. Protein amounts in lysates were determined by Pierce BCA protein assay kit (Thermo Fisher Scientific). After SDS-PAGE, proteins were transferred onto PVDF membranes (Bio-Rad). Membranes were blocked with 5% skim milk in Tris-buffered saline containing 0.1% Tween-20. After blocking, membranes were incubated with primary antibody (listed below) for overnight at 4 °C. Membranes were incubated with secondary antibody (listed below) for 1 h at RT. Washed membranes were analyzed with EZ-Western Lumi Pico or Femto reagent (DoGen). Band intensity was measured by Fusion-Solo, and gated band intensities from acquired images were analyzed by Fusion-capt advance (Vilber Lourmat). Mouse anti-Chi3l1 antibody was from R&D Systems (1:1000 diluted, MAB2649), and mouse p-Erk (1:1000 diluted, #4377S), p-Akt (1:1000 diluted, #4060S), pSTAT1 (1:1000 diluted, #7649), pSTAT4 (1:1000 diluted, 4134S), pSTAT6 (1:1000 diluted, #9361 P), tSTAT1 (1:1000 diluted, #9172 P), tSTAT4 (1:1000 diluted, 2653S), and tSTAT6 (1:1000 diluted, #9362 P) antibodies and HRP-conjugated anti-rabbit (1:20000 diluted, #7074) were from Cell Signaling Technology. HRP-conjugated anti-rat (1:10000 diluted, 405405) antibody was from Biolegend. Raw western blotting images are presented in Supplementary Fig 16.

**Quantitative RT-PCR**. Total RNA from in vitro differentiated CD4 T cells, CD8 T cells, and separated lung lymphocytes were isolated with RNeasy Mini kits (Qiagen) following the manufacturer's protocol. Lung tissue was disrupted by a homogenizer with TRIzol (Thermo Fisher Scientific), and total RNA was extracted. cDNA was synthesized with ReverTra Ace qPCR RT master mix (Toyobo). RT-PCR was performed on a Bio-Rad CFX Connect RT-PCR detection system using iQ SYBR Green Supermix (Bio-Rad). Primer sequences used are listed in Supplementary Table 1.

**RNA-sequencing**. Total RNA was isolated from WT and Chi3l1 KO naïve and Th1 cells using TRIzol reagent (Invitrogen). RNA quality was assessed by Agilent 2100 bioanalyzer using RNA 6000 Nano Chips (Agilent Technologies, Amstelveen, Netherlands). RNA quantification was performed using an ND-2000 spectrophotometer (Thermo Fisher Scientific). For control and test RNAs, library construction was performed using SENSE mRNA-Seq Library Prep Kits (Lexogen, Inc., Austria) according to the manufacturer's instructions. Briefly, 2 μg of total RNA is prepared and incubated with magnetic beads decorated with oligo-dT, then all other RNAs except mRNA were removed by washing. Library production is initiated by random hybridization of starter/stopper heterodimers to the poly(A) RNA still bound to the magnetic beads. These starter/stopper heterodimers contain Illumina-compatible linker sequences. A single-tube reverse transcription and ligation reaction extends the starter to the next hybridized heterodimer, where the newly-synthesized cDNA insert is ligated to the stopper. Second-strand synthesis is performed to release the library from the beads, and the library is then amplified. Barcodes were introduced when the library was amplified. High-throughput sequencing was performed as paired-end 100 sequencing using a HiSeq 2000 instrument (Illumina, Inc., USA). mRNA-Seq reads were mapped using the TopHat software tool to obtain an alignment file. Differentially expressed genes were determined based on counts from unique and multiple alignments using Edge R within R version 3.2.2 (R development Core Team, 2011) using BIO-CONDUCTOR version 3.0. The alignment file was used for assembling transcripts, estimating abundances, and detecting differential expression of genes or isoforms using cufflinks. FPKM (fragments per kilo base of exon per million fragments)

method used to determine expression levels of gene regions. A global normalization method was used for comparisons between samples. Gene classification was based on searches performed by DAVID (http://david.abcc.ncifcrf.gov/).

**dNP2-siRNA complex formation**. To determine optimal conditions for forming cell-permeable peptide-siRNA complexes, EGFP or Chi3l1 siRNA (Bioneer) was mixed with dNP2-HA2 peptide (Genscript, KIKKVKKKGRKGSKIKKVKKKGRK-GLFGAIAGFIENGWEGMIDG) which is a combined sequence of a cell-penetrating peptide dNP2 and endosomal escape sequence of HA2[22] at various N/P ratios (the number of amine groups in dNP2-HA2 peptide to the number of phosphate groups in siRNA) and incubated for 30 min at RT in the dark. The size of dNP2-siRNA complex was measured by Nano particle size analyzer (K-one).

**Gel retardation assay**. 1.5 µg of Chi3l1 siRNA (Bioneer) was mixed with dNP2-HA2 peptide at various N/P ratios and incubated for 30 min at RT in the dark. After incubation, complexes were mixed with DNA loading dye, and gel electro-phoresis was performed on 2% agarose gel for 20 min.

**Pulmonary melanoma metastasis model**. The mouse melanoma cell line (B16F10) established from C57BL6/J mouse melanoma was purchased from ATCC (CRL-6475). B16F10 cells are maintained in complete DMEM at 37 °C. After cul-turing to 90% confluence in complete DMEM, cells were harvested, adjusted to $10^6$ cells/mL in pre-warmed PBS, and $5 \times 10^5$ cells were i.v. injected into C57BL6/J, Chi3l1 KO, CD4-specific Chi3l1 KO, and RAG KO mice. Mycoplasma con-tamination was checked before the injection. RAG KO mice had transferred in vitro differentiated WT or Chi3l1 KO Th1/CTL by i.v. injection at 5 days after melanoma injection. At day 14, mice were sacrificed, and the numbers of melanoma colonies visualized as black dots on the lung surface were counted. For histological analysis, paraffin blocks of lung tissues were deparaffinized and stained by hematoxylin and eosin. Tumor sizes in lung tissue were measured by Image J software 1.48 v. Lung tissues were mechanically chopped and incubated in Ca2+-containing and Mg2+-containing DPBS with collagenase D (1 mg/mL, Roche) for 30 min at 37 °C. In experiment for Supplementary Fig. 3, tumor region and non-tumor region were mechanically separated, and digested by collagenase D. Enzymatic digestions were stopped with 0.5 M EDTA solution. Digested cells were filtered by 40-µm cell strainer, and RBCs were lysed with Ack buffer. In addition, to confirm systemic effect of Chi3l1 KO or dNP2-siChi3l1 treatment, splenocytes and inguinal lymph node cells were analyzed. Lung lymphocyte cells were enriched by Percoll gradients, and the cells were analyzed by flow cytometry and quantitative RT-PCR.

**Subcutaneous tumor growth model**. $1 \times 10^5$ of B16F10 melanoma cells were s.c. injected into C57BL6/J, CD4-Chi3l1 KO mice. The size of tumor was checked for every 2 days. At day 18, mice were sacrificed, and the tumor tissue was dissected for further experiment. Tumor sizes were measured by caliper. Tumor tissues were mechanically chopped and incubated in Ca2+-containing and Mg2+-containing DPBS with collagenase D (1 mg/mL, Roche) for 30 min at 37 °C. Enzymatic digestions were stopped with 0.5 M EDTA solution. Digested cells were filtered by 40-µm cell strainer, and RBCs were lysed with Ack buffer. Tumor infiltrated lymphocyte cells were enriched by Percoll gradients, and the cells were analyzed by flow cytometry.

**In vitro cytotoxicity assay**. B16F10 melanoma cells were seeded into 96-well plates in 100 µL media. After WT and Chi3l1 KO FACS-sorted CD8 T cells were activated with plate-bound anti-CD3 and anti-CD28 antibodies for 4 h, the cells were added to wells containing B16F10 cells at different effector to target (E:T) ratio. For NK cell cytotoxicity assay, FACS-sorted NK cells were activated with 1000 U/mL IL-2 for 10 days, then the cells were re-activated with plate-bound anti-NKG2D antibody (eBioscience, 16-5872-82) for 4 h. The pre-activated NK cells were added to wells containing B16F10 cells at different effector to target (E:T) ratio. Remained live cells were stained with CCK-8 and the percent CTL and NK cell activity was calculated as relative percentage to the OD value of B16F10 only control.

**In vitro dNP2-siRNA transfection**. To express Chi3l1 in HEK293T/17 cells (ATCC, CRL-11268), cells were transfected with a cloned Chi3l1-overexpression vector using Lipofectamine 2000 (Invitrogen). Transfection followed the manu-facturer's protocol. To make dNP2-siChi3l1 complex, indicated concentration of siChi3l1 was mixed with dNP2-HA2 peptide. Mixture was incubated for 30 min at RT in the dark. dNP2-siChi3l1 was added independently, and media were replaced 6 h later. Transfected cells were incubated for 2 days, and media and cells were harvested. Chi3l1 levels in culture media were determined by ELISA (R&D), and Chi3l1 mRNA in transfected cells was quantified by RT-PCR.

**In vivo dNP2-siRNA transfection**. 1.25 or 2.5 µg of Chi3l1 siRNA (Bioneer) or EGFP siRNA (Bioneer) was incubated with 35 or 70 µg dNP2-HA2 peptide for 30 min at RT, in the dark. Free siRNA or Peptide-siRNA complexes were intranasally administered to mice, which were sacrificed every 24 h for 3 days. Lung tissue was disrupted in T-PER Tissue Protein Extraction Reagent (Thermo Fisher Scientific) for protein lysates or in TRIzol for RNA extraction. Chi3l1 protein in the lung was

measured by Chi3l1 ELISA kit (R&D Systems). In the lung metastasis model, $5 \times 10^5$ B16F10 cells were intravenously injected to WT mice, and then 2.5 µg of siRNA + 70 µg of dNP2-HA2 peptide complex was intranasally administered to anesthetized mice with zoletil + Rompun for twice per day with 10 h interval for every other day until day 14. At 14 days after melanoma injection, mice were sacrificed and analyzed.

**Statistical analysis**. All data were analyzed by two-tailed Student's *t*-test using Prism5 (GraphPad). *P*-values less than 0.05 were considered statistically significant.

**Data availability**. The total RNA sequencing data have been deposited in the GEO database under the accession code GSE105056.The authors declare that all the other data supporting the findings of this study are available within the article and its Supplementary Information Files, or from the corresponding authors on reasonable request.

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

## Acknowledgements

This research was supported by the Basic Science Research Program (NRF-2013R1A1A2A10060048) and the Bio and Medical Technology Development Program (NRF-2017M3A9C8027972) of the NRF funded by the Korean government. This work was also in part supported by grant PO1 HL114501, and R01 HL115813 from National Institute of Health (NIH), USA. We also thank Yeon-Ho Kim for technical support of FACS sorting at Hanyang LINC analytical Equipment Center (Seoul).

## Author contributions

J. -M. C. conceived and supervised the study; D. -H. K. designed and performed most of the experiments. H. -J. P., S. L., J. -H. K., H. -G. L., J. O. C. and J. H. O. supported experiments; D. -H. K., H. -J. P., S. L., C. -G. L., J. A. E. and J. -M. C. analyzed and discussed the data; S. -J. H., M. -J. K., C. M. L., C. -G. L. and J. A. E. provided animals and reagents; D. -H. K. and J. -M. C. wrote the manuscript; All authors edited the manuscript.

## Additional information

**Competing interests:** D. -H. K., and J. -M. C. are inventors on a patent application describing the use of dNP2-siRNA complex in cancer immunotherapies. The remaining authors declare no competing financial interests.

