## [Peer Review File · Nature Communications]

Reviewer #2 (Remarks to the Author):

The authors describe the function of Chi3l1 in T cell responses in the context of cancer. Chi3l1 appears to be a negative regulator of CD4 and CD8 T cells, wherein KO cells stimulated by antiCD3/antiCD28 produce increased IFN γ compared to WT cells. This process is mediated by regulation of IFN γ -mediated STAT1 phosphorylation. Chi3l1 KO Th1 cells are characterized by a distinct transcriptional program, with decreased twist1 and socs1. Chi3l1 KO mice challenged with B16F10 melanoma cells have fewer lung metastasis and increased IFN γ + CD4/CD8 cells, without impact on KN cells. In vivo inhibition of Chi3l1 results in significantly fewer lung metastasis and increased T cell infiltration and activation, proposing that targeting Chi3l1 may be effective to enhance anti-tumor immunity.

Overall, the experiments seem thoughtfully considered and the multi-part story well constructed. There are some places where mechanistic links are not completely demonstrated and where the data is described in a way that makes it challenging to know if there is reliable/biologically meaningful differences.

Figure 1:

- It is noted that expression of Chi3l1 is particularly high "among the chitinases" in CD4 and CD8 cells; and Fig 1A and B depicts "relative expression." Can comment be made about the absolute degree of expression of Chi3l1? It would facilitate understanding of how substantial Chi3l1 expression is in T cells more generally than in the specific context of chitinases. Also, is Chi3l1 expression increased in WT CD4 and 8 cells stimulated by antiCD3/antiCD28, which could help highlight the functional role in restraining T cell activation.

- Similarly, despite the statistical significance of the changes in IFN γ in Chi3l1 KO cells stimulated by antiCD3/antiCD28, some of the absolute differences seem small (e.g. IFN γ ~2ng/mL to ~3.5ng/mL in CD4s; compared to ~60ng/mL to ~110ng/mL in CD8s). Can comment be made about which changes are likely to be biologically significant, or help contextualize the degree of differences seen?

- It is demonstrated that loss of expression increased T cell activation and proliferation. Does increased Chi3l1 (e.g. HEK293T overexpressing cells used in Fig 5) restrain T cell activation?

Figure 3:

- Instead of 3C, a volcano plot of the RNAseq data depicting p-value and fold change expression in the Th1 WT vs KO cells would be more convincing of meaningful gene expression differences.

- In the quantitative rtPCR data, are the directionality and degree of differences in gene expression consistent across the 3 experiments? This would help assess the reliability of the findings.

Figure 4:

- To further confirm the T cell-mediated mechanism, if T cells are inhibited in Chi3l1 KO and WT models, is difference in lung metastasis following melanoma challenge lost?

Figure 5 and 6:

- The degree and sustainedness of Chi3l1 decreased expression seems relatively modest. The changes in IFN γ alone (rather focus on IFN γ +TNF α + double positive cells) also seem modest, if at all (e.g. mean of IFN γ + cells in dNP2siEGFP CD8s is 47.8% vs 53.7% in 250ng dNP2-siChi3l1). It is somewhat surprising, then, that the degree of decrease in lung metastases is seen in vivo in Fig 6. Is it possible there is a direct effect of the siRNA construct on melanoma (e.g. is Chi3l1 expressed in melanoma cells)? Similar to comment in Fig 4, can it be confirmed that the phenotypic change in the context of the siRNA construct is indeed T cell mediated?

Lastly, to highlight how relevant this pathway is in human cancers, is there publicly available data (e.g. RNAseq data from TCGA or otherwise) that could be examined to characterize the frequency with which Chi3l1 is dysregulated in the microenvironment of cancers?

Reviewer #3 (Remarks to the Author):

Chitinase-3-like protein 1 (CHI3L1), is a glycoprotein expressed and secreted by various cell-types, including macrophages, fibroblast-like synovial cells, vascular smooth muscle cells, and hepatic stellate cells. CHI3L1 expression has been associated with pathogenic processes related to inflammation, extracellular matrix remodeling, asthma, and solid carcinomas and GBM. However, the biological function and molecular mechanism of action remain unclear.

In this manuscript the authors focus on the cell-autonomous role of CHI3L1 in cytotoxic T-cell. They propose CHI3L1 as therapeutic target to prevent lung metastases.

Following the breakthrough of cancer immunotherapy in the last decade, there is growing interest around new strategies that could target the immune system and improve the response of patients to treatments. In this scenario, this work could be of certain interest, also considering the innovative in vivo therapeutic approach presented here. However, the evidences are mostly indirect, the

molecular mechanism is not clearly addressed, and the paper lacks conclusive in vivo experiments. The paper could be significantly improved by addressing the following concerns.

Major concerns:

- All the in vivo experiments should be repeated both in a Rag KO mouse model and in an orthotropic model of tumor metastasis, to confirm that the observed phenotype is truly dependent from T-cell and Infg (as stated in the title), and to address the effect on the primary tumor growth (claimed in the abstract), respectively.

-Because of the hyper activation of the CTL compartment upon CHI3L KO and because of published data showing the involvement of CHI3L1 in GvHD, adoptive transfer experiments are required to address the the Graft vs Tumor (GvT)/GvHD benefits. This would give a better idea of the translational potential of CHI3L1 as target for immunotherapy.

-The effect of the in vivo genetic (and not chemical) CHI3L1 inhibition are not conclusive. The authors should provide data supporting the KO is really affecting the T-cell function in vivo and that the proposed effect on tumor lung seeding is not mediated by other cell compartments.

Other comments:

-Characterization of tumor infiltrates should include also quantitative data (i.e. CD4 and CD8 inside CD3), both by flow and IHC. This is important also considering that any difference of CTL activity in the in vitro assay disappears when E:T ratio doubles.

-There is no clear understanding of the mechanism. For example, how the intracellular CHI3L1 differs from the secreted one? also, data about possible downstream regulators are only correlative and need to be confirmed in loss-of-function experiments.

-Tumor specific T-cells should be compared with T-cell in the periphery upon in vivo siRNA

-Due to the slight changes when comparing mRNA and protein levels, all the bar graphs should be replaced with scatter plot, showing the single values. This would exclude the possible effect of outliers.

Reviewers' comments:

Reviewer #2 (Remarks to the Author):

The authors describe the function of Chi3l1 in T cell responses in the context of cancer. Chi3l1 appears to be a negative regulator of CD4 and CD8 T cells, wherein KO cells stimulated by antiCD3/antiCD28 produce increased IFN γ compared to WT cells. This process is mediated by regulation of IFN γ -mediated STAT1 phosphorylation. Chi3l1 KO Th1 cells are characterized by a distinct transcriptional program, with decreased twist1 and sox1. Chi3l1 KO mice challenged with B16F10 melanoma cells have fewer lung metastasis and increased IFN γ +CD4/CD8 cells, without impact on KN cells. In vivo inhibition of Chi3l1 results in significantly fewer lung metastasis and increased T cell infiltration and activation, proposing that targeting Chi3l1 may be effective to enhance anti-tumor immunity.

Overall, the experiments seem thoughtfully considered and the multi-part story well constructed. There are some places where mechanistic links are not completely demonstrated and where the data is described in a way that makes it challenging to know if there is reliable/biologically meaningful differences.

Answer; We would like to thank to this reviewer with positive opinion on our study. We pleasantly can address all this reviewer's questions to improve reliability of biologically meaningful differences.

Figure 1:

- It is noted that expression of Chi3l1 is particularly high "among the chitinases" in CD4 and CD8 cells; and Fig 1A and B depicts "relative expression." Can comment be made about the absolute degree of expression of Chi3l1? It would facilitate understanding of how substantial Chi3l1 expression is in T cells more generally than in the specific context of chitinases. Also, is Chi3l1 expression increased in WT CD4 and 8 cells stimulated by antiCD3/antiCD28, which could help highlight the functional role in restraining T cell activation.

Answer; As the reviewer requested, we revised the demonstration of the data as relative expression of delta-Ct value to β -actin which would be an available way to show absolute degree of expression of Chi3l1 in T cells. To address its significance of the expression level in T cells, we preformed western blot analysis of its protein level in activated T cells which confirms significant protein level of Chi3l1 was detected following TcR stimulation. As reviewer commented, increase of Chi3l1 expression results could be useful to speculate its role in T cell activation. In addition, we examined Chi3l1 expression in Th0,1,2,17, Treg subsets emphasizing its possible role in regulation of Th1 vs Th2 differentiation. We replaced the figure 1 with dCT value and western blot results. We also replaced supplementary figure 11 to show Chi3l1 expression level in effector T cell subsets with revised text in the manuscript.

Replaced Fig. 1

Replaced supplementary Fig. 11

- Similarly, despite the statistical significance of the changes in IFN γ in Chi311 KO cells stimulated by antiCD3/antiCD28, some of the absolute differences seem small (e.g. IFN γ ~2ng/mL to ~3.5ng/mL in CD4s; compared to ~60ng/mL to ~110ng/mL in CD8s). Can comment be made about which changes are likely to be biologically significant, or help contextualize the degree of differences seen?

Answer; When sorted naïve CD4 T cells were stimulated with anti-CD3/CD28 antibodies (Th0 condition), IFN γ in culture supernatant level is apparently increasing but much lower than Th1 skewed cells like in Fig.2. (1.801ng/mL in Th0 vs 26.34ng/mL in Th1 cells). It is significantly different between WT and Chi311 KO CD4 T cells while the amount of IFN γ produced by CD8 T cells are much higher. This initial observation prompted us to further investigate IFN γ levels in Th1 WT and KO showing much significant difference of IFN γ production. We can add sentences in the text to better contextualize the meaning of initial observation of IFN γ level in Fig. 1.

- It is demonstrated that loss of expression increased T cell activation and proliferation. Does increased Chi311 (e.g. HEK293T overexpressing cells used in Fig 5) restrain T cell activation?

Answer; Chi311 has been studied as its soluble protein in the blood to affect various cell types. This question seems to concern rescue experiments whether Chi311 overexpression could reverse the phenotype. First of all, we attempted to treat 1-5 μ g of commercial recombinant Chi311 protein to T cells and macrophages. As shown as follows and new supplementary figure 9, recombinant protein could not affect CD4 T cell differentiation however it inhibited cytokine production in both wild type and knock out macrophages suggesting its intrinsic role in T cells. To further address this question, we can utilize retro-virus vector for expressing Chi311 in T cells. We already made virus particle to transfect into the T cells and hope to get a result soon.

Figure 3:

- Instead of 3C, a volcano plot of the RNAseq data depicting p-value and fold change expression in the Th1 WT vs KO cells would be more convincing of meaningful gene expression differences.

Answer; We can re-analyze it as volcano plot if this is necessary.

- In the quantitative rtPCR data, are the directionality and degree of differences in gene expression consistent across the 3 experiments? This would help assess the reliability of the findings.

Answer; Yes, we analyzed RT-PCR data with at least three independent experiments. We added words for clarify this in the figure legend.

Figure 4:

- To further confirm the T cell-mediated mechanism, if T cells are inhibited in Chi311 KO and WT models, is difference in lung metastasis following melanoma challenge lost?

Answer; This question is based on wondering KO phenotype is T cell specific. The suggested experiment by the reviewer could be T cell depletion experiments by antibody treatment. Direct experimental design, however, would be utilizing T cell specific Chi311 knock out mice. To address this question, we firstly generated CD4-Chi311 KO mice, which was not reported yet, to confirm that Chi311 deletion in T cells increases anti-tumor immunity as shown in following data. In consistent with observed phenotypes in total KO mice, we found the number of tumor nodules growing in the lung is significantly reduced with increase of IFN γ and TNF α producing both CD4 and CD8 T cells but not NK or non-lymphocytes in the lung suggesting inhibition of melanoma metastasis is pretty much T cell-mediated. We also analyzed tissue histology and RT-PCR which was added as a new Fig. 5.

Figure 5 and 6:

- The degree and sustainedness of Chi311 decreased expression seems relatively modest. The changes in IFN γ alone (rather focus on IFN γ +TNF α + double positive cells) also seem modest, if at all (e.g. mean of IFN γ + cells in dNP2siEGFP CD8s is 47.8% vs 53.7% in 250ng dNP2-siChi311). It is somewhat surprising, then, that the degree of decrease in lung metastases is seen in vivo in Fig 6. Is it possible there is a direct effect of the siRNA construct on melanoma (e.g. is Chi311 expressed in melanoma cells?)? Similar to comment in Fig 4, can it be confirmed that the phenotypic change in the context of the siRNA construct is indeed T cell mediated?

Answer; Logical strategy to study about Chi311 in T cells is based on phenotypic analysis of Chi311 KO T cells and lung metastasis model. This is one important innovation of the study that Chi311 is claimed as a novel molecule for controlling Th1 and CTL responses. We have shown it in vitro and in vivo by utilizing Chi311 total knock out and T cell specific knock out mice model. Then, finally, we would like to prove whether Chi311 is truly important target for enhancing tumor immunity for rejecting tumors. One time treatment of dNP2-siChi311 reduces Chi311 mRNA level in the lung at day-2 the most then start to be recovered. This observation is pretty much identical with Dr. SF. Dowdy's siRNA in vivo delivery paper which is one of pioneer paper regarding siRNA delivery in vivo published in Nature Biotechnology 2009. We reported better cell-penetrating peptide, dNP2, which could deliver a protein into T cells while TAT could not in Nature Communications 2015. Here we firstly attempted siRNA delivery with dNP2. In vivo knockdown of Chi311 mRNA and protein level is apparently significant compare to control group of dNP2-siEGFP treatment. Another innovative point of our study is we could successfully control T cell responses by siRNA delivery into T cells. Th1 skewed cells treated with dNP2-siChi311 showing significant increase of IFN γ producing cells which is very consistent result of Chi311 KO T cells. However, as the reviewer doubted the result of CD8 T cells, IFN γ + cells in dNP2-siEGFP of CD8 T cells is 32.812% vs 41.56% in 250 ng dNP2-siChi311. This possibly be due to CD8 T cells are more sensitively activated compare to CD4 T cells which might little overcome siRNA effect and it may not be optimal in vitro condition. Therefore, we repeated this experiments and replaced the CD8 T cell data in Fig. 6h.

In addition, as the reviewer raised the issue whether dNP2-siChi311 directly inhibits B16F10 cell growth in vivo, we treated 100ng, 250ng of dNP2-siChi311 onto melanoma cells, then analyzed the proliferation as shown in following data which was added as a new supplementary figure 12. The result demonstrates there is no effect of siRNA treatment on tumor cell growth suggesting there would be no significant effect on tumor cells. In addition, Chi311 expression level is just little in B16F10 compare to peritoneal macrophage, EL4, RAW264.7 cells.

As we addressed in above to this reviewer's question regarding T cell specificity of Chi311 KO phenotype, in vivo prevention of lung melanoma metastasis is mostly T cell mediated. In addition, we showed NK cells or other cell types were not expressing IFN γ in the lung by dNP2-siChi311 treatment. However, dNP2-siChi311 treatment in vivo would not still reasonable to conclude it is only T cell specifically working for rejecting tumors due to the delivery efficiency of dNP2 peptide. dNP2 could deliver a molecule into various cells which would not be limited to T cells. Therefore, to address dNP2-siChi311 in other cell types, we performed in vitro experiments of the treatment in peritoneal macrophages showing that they could induce more IL-12 and IL-6 upon LPS stimulation suggesting decrease of Chi311 in macrophage also could indirectly support Th1-favorable environment. Thus, we think in vivo treatment of dNP2-siChi311 synergistically working on macrophage and T cells to enhance Th1 immunity still not much on NK cells due to the less expression of Chi311. We added following data as a new supplementary figure 13.

Lastly, to highlight how relevant this pathway is in human cancers, is there publicly available data (e.g. RNAseq data from TCGA or otherwise) that could be examined to characterize the frequency with which Chi311 is dysregulated in the microenvironment of cancers?

Answer; This question is about human relevance. To address this question, we attempted to treat dNP2-siChi311 in human PBMC that it consistently increased IFN γ production by human CD4 and CD8 T cells as shown in the following data which was added as a new supplementary figure 14.

As also reviewer requested we will attempt to access TCGA for RNAseq analysis. This question would be a valuable subject in discussion section. We actually would like to emphasize possible intrinsic role of Chi311 in T cells in our study. Various studies suggested that Chi311 is a biomarker of types of tumor (Oncotarget. 2017 Jan 17;8(3):5382-5391., Diagn Pathol. 2016 Apr 27;11:42., Melanoma Res. 2016 Aug;26(4):367-76.). Increased level of Chi311 was observed in human breast cancer patients with poor prognosis, and it involved to progression of astrocytoma and poor patient survival for glioblastoma and lower grade astrocytoma tumor. Recently, N Cohen et al. (2017 Apr 3.) suggested fibroblast cells support Chi311 in tumor microenvironment for suppressing anti-tumor immunity which supports our hypothesis of the direction. The authors showed that increased Chi311 by fibroblast enhances migration and growth of breast cancer, and knockdown of Chi311 by shRNA increases T cell population and IFN γ , TNF α expression. We will add this reference and make a paragraph for better discussion.

Reviewer #3 (Remarks to the Author):

Chitinase-3-like protein 1 (CHI3L1), is a glycoprotein expressed and secreted by various cell-types, including macrophages, fibroblast-like synovial cells, vascular smooth muscle cells, and hepatic stellate cells. CHI3L1 expression has been associated with pathogenic processes related to inflammation, extracellular matrix remodeling, asthma, and solid carcinomas and GBM. However, the biological function and molecular mechanism of action remain unclear.

In this manuscript, the authors focus on the cell-autonomous role of CHI3L1 in cytotoxic T-cell. They propose CHI3L1 as therapeutic target to prevent lung metastases.

Following the breakthrough of cancer immunotherapy in the last decade, there is growing interest around new strategies that could target the immune system and improve the response of patients to treatments. In this scenario, this work could be of certain interest, also considering the innovative *in vivo* therapeutic approach presented here. However, the evidences are mostly indirect, the molecular mechanism is not clearly addressed, and the paper lacks conclusive *in vivo* experiments. The paper could be significantly improved by addressing the following concerns.

Answer; We would like to thank to this reviewer for valuable critiques on our study. We pleasantly can address all questions to improve direct evidence on regulating T cell immunity *in vivo* with better mechanisms.

Major concerns:

- All the *in vivo* experiments should be repeated both in a Rag KO mouse model and in an orthotropic model of tumor metastasis, to confirm that the observed phenotype is truly dependent from T-cell and *Irfng* (as stated in the title), and to address the effect on the primary tumor growth (claimed in the abstract), respectively.

Answer; This question claims whether reduced lung metastasis is T cell dependent *in vivo*. To directly address the question, we generated T cell specific Chi311 knock out mice by crossing CD4-Cre mice with Chi311 fl/fl mice and analyzed B16F10 lung metastasis. As shown in the following data, it was successfully repeated as consistent results that melanoma lung metastasis is significantly reduced in CD4-Chi311 KO mice with increase of IFN γ producing both CD4 and CD8 T cell infiltration in the lung. We also analyzed tissue histology and RT-PCR to confirm the phenotype shown in total knock out mice which was added as a new Fig. 5.

As the reviewer suggested, we also attempted T cell transfer model in RAG KO mice to double-check its role in T cells. We transferred 2×10^5 Th1 and 3×10^5 activated CD8 T cells after 5×10^5 B16F10 melanoma cell transfer into RAG KO mice. As shown in the data, wild type Th1 and CTL transfer could significantly reduced number of pleural colonies with increase of CD4 and CD8 T cells in the lung. Chi311 KO T cells more significantly reduced the metastasis phenotype even compare to wild type T cell transfer result. More significantly, increase of IFN γ producing cells were observed in Chi311 KO T cell transferred RAG KO mice collectively suggesting inhibition of lung metastasis and increase of tumor immunity is truly due to the lack of Chi311 in T cells. This result was added as a new supplementary figure 6.

We also attempted orthotopic model of tumor growth by injecting B16 melanoma cells in subcutaneously. Although the tumor size was not significantly affected in preliminary experiment, IFN γ producing T cells in the tumors were significantly increased. We will further perform the experiments to get solid conclusion during revision period.

For better precise description of our findings, we revised abstract sentences as “inhibit pulmonary metastasis” from “tumor growth and progression”. And also revised Introduction sentences as “these studies suggest that Chi311 plays an essential role in regulation of Th1 and CTL differentiation. These studies highlight that specific intervention of Chi311 in T cells could be an effective therapy of pulmonary metastasis.” as highlighted yellow from “These studies highlight that specific intervention of Chi311 could be an effective therapy of the cancers in which Chi311 is significantly dysregulated.”

In addition, we attempted dNP2-siChi31 treatment after day-2 of melanoma cell transfer into the mice which is the setting of the experiments whether dNP-siChi31 treatment could inhibit tumor growth. As shown in the following data, melanoma lung metastasis was dramatically reduced by dNP2-siChi31 treatment even after day-2 with increase of IFN γ and TNF α producing T cells suggesting it could control tumor growth as well as metastasis. We added this figure as a new supplementary figure 8.

-Because of the hyper activation of the CTL compartment upon CHI3L KO and because of published data showing the involvement of CHI3L1 in GvHD, adoptive transfer experiments are required to address the the Graft vs Tumor (GvT)/GvHD benefits. This would give a better idea of the translational potential of CHI3L1 as target for immunotherapy.

Answer; One recent paper as the reviewer mentioned suggested that Chi31 deficiency in donor T cells increased the severity of acute GVHD through enhancing inflammation, T cell proliferation, and production of Th-related cytokines. Transfer of T-cell depleted BM into irradiated BALB/c mice with C57BL/6 background Chi31 deficient splenocytes induces more severe GvHD compare to WT splenocytes. 14 days later from transfer, splenocytes from recipient mouse were analyzed by flow cytometry. Increased CD3 T cell proliferation and up-regulated IFN γ , TNF α in CD4 and CD8 T cells were observed in Chi31 deficient splenocytes transferred mice, suggesting Chi31 suppresses IFN γ and TNF α expression by CD4, CD8 T cells in GvHD models. It would be interesting idea to study GvT/GvHD as a separate story. We added this reference in discussion section.

-The effect of the in vivo genetic (and not chemical) CHI3L1 inhibition are not conclusive. The authors should

provide data supporting the KO is really affecting the T-cell function in vivo and that the proposed effect on tumor lung seeding is not mediated by other cell compartments.

Answer; For addressing whether Chi3l1 KO in vivo phenotype is truly depends on altering T cell function, as we addressed above, we confirmed it in CD4-Chi3l1 KO mice and RAG KO mice T cell experiment. In original version of the figure already showed increase of IFN γ and anti-tumoral molecules are not produced by NK cells or non-lymphocytes. For siRNA treatment experiment, however, in vivo genetic knock down by dNP2-siChi3l1 would affect other cell types as well. To address how dNP2-siChi3l1 would affect to decrease of lung metastasis, we attempted to treat dNP2-siChi3l1 in peritoneal macrophage in vitro and stimulated them with LPS. As shown in the data, reduced Chi3l1 expression in macrophage increases IL-12 and IL-6 suggesting it possibly support Th1 differentiation in vivo tumor environment. In summary, we confirmed T cell mediated function in vivo using CD4-Chi3l1 KO mice with melanoma metastasis and dNP2-siChi3l1 would affect macrophages which might synergistically induces more Th1 and CTLs in vivo to control tumor growth or metastasis. We added following data as a new supplementary figure 13.

Other comments:

-Characterization of tumor infiltrates should include also quantitative data (i.e. CD4 and CD8 inside CD3), both by flow and IHC. This is important also considering that any difference of CTL activity in the in vitro assay disappears when E:T ratio doubles.

Answer; Tumor infiltrating T cells are important to reject tumors as reviewer commented. As the reviewer requested, we can analyze tumor infiltrating T cells by FACS and IHC. At first, by flowcytometry, we separately collected TIL and Non TIL from tumor and normal tissue of wild type and CD4-specific chi311 KO mice. As shown in the new supplementary figure 5, tumor-infiltrating CD45 positive cells were increased in CD4-Chi311 KO mice, however no difference in non-tumor region. Interestingly CD4 T cells in CD45 positive cells were increased in both non-tumor and tumor region. More interestingly IFN γ producing CD4 and CD8 T cells in CD4-Chi311 KO mice were increased in the tumor whereas only IFN γ producing CD4 T cells were significantly increased in non-tumor region. Because tumor region of CD4-Chi311 KO mice are very small and limited, we pooled 5 mice samples for TIL FACS. We can confirm this by performing additional independent experiment.

-There is no clear understanding of the mechanism. For example, how the intracellular CHI3L1 differs from the secreted one? also, data about possible downstream regulators are only correlative and need to be confirmed in loss-of-function experiments.

Answer; We agree that intrinsic role of Chi3l1 with molecular mechanism how to control Th1/2 balance would be important question. Previously, secreted Chi3l1 was found in the serum of various inflammatory diseases or cancers however there was only one report (Immunobiology. 2016 Feb;221(2):347-56.) which showed Chi3l1 localization in nucleus and its role in monocyte-derived DC maturation, suggesting possible intrinsic role of Chi3l1 in the cells. This is one of innovative finding of our work suggesting Chi3l1 is important cell-intrinsic regulator of T cell immune response. IL-13Ra2 and TMEM-219 was suggested as receptor compartments for extrinsic Chi3l1 recognition and signaling pathway including Erk/Akt phosphorylation, β -catenin translocation (Cell Rep. 2013 Aug 29;4(4):830-41., Nat Commun. 2016 Sep 15;7:12752.), however, we found that IL-13Ra2 is not expressed in both naïve or activated T cells. When we treat commercial recombinant Chi3l1 recombinant protein onto T cells and macrophages, interestingly, only macrophage function was inhibited by exogenous Chi3l1 protein treatment suggesting altering T cell function would be rely on more intrinsic based mechanism.

For speculate molecular mechanisms, based on RNAseq analysis, we proposed T-bet, Twist1, SOCS1 would be one of important target to regulate IFN γ signaling and Th1 differentiation which was affected by loss of Chi3l1 in T cells. Our hypothesis is reduced Chi3l1 decreases Twist1 expression which is a negative regulator of Th1 differentiation and reported to regulate T-bet and SOCS1 expression (J Exp Med, 205, 1889-1901 (2008)) results robust Th1 differentiation. We thought detail molecular mechanism would be a next story, however, to more clearly address molecular mechanisms, we are attempting Co-IP experiments whether Chi3l1 physically interacts with Twist-1 or other target molecules.

-Tumor specific T-cells should be compared with T-cell in the periphery upon in vivo siRNA

Answer; This would be also important question regarding circulating T cells vs tumor-infiltrating T cells in vivo upon dNP2-siChi31 treatment. Frankly, we already checked it when we performed the experiments described in original Fig. 5. It shows splenic or inguinal lymph node T cells in dNP2-siChi31 treated mice does not have any significant differences in the context of TNF α and IFN γ production. Additionally, we checked those peripheral cells in CD4-Chi31 KO mice with melanoma cells that there is no difference neither. These data suggesting that nasally administrated dNP2-siChi31 only affect to T cells in the lung rather T cells in the periphery. We added this figure as a new supplementary figure 4.

-Due to the slight changes when comparing mRNA and protein levels, all the bar graphs should be replaced with scatter plot, showing the single values. This would exclude the possible effect of outliers.

Answer; As the reviewer suggested, we replaced the bar graph as scatter plot.

Reviewer #2:

Remarks to the Author:

Overall it seems like the authors have reasonably responded to the comments from reviewers. I don't have additional comments.

Reviewer #4:

Remarks to the Author:

In the present manuscript "Inhibition of Chitinase 3-like-1 expression enhances Th1 and CTL function via the IFN γ /STAT1 axis to prevent lung metastasis", the authors demonstrate a novel immune regulatory function of Chi3l1 in T cells and suggest that Chi3l1 constitutes an effective therapeutic target to prevent pulmonary metastasis.

Although the work is of a major interest in the field of cancer immunotherapy, several points need to be addressed and protocol to be clarified.

Comments

The authors previously reported a new cell penetrating peptide dNP2 isolated from the human novel LZAP-binding protein. The tandem peptide dNP2 has been successfully applied to deliver an immune regulatory protein into T cells and in vivo. In the present work, they applied dNP2 peptide to improve the delivery of siRNA targeting Chi3l1 into both cultured cells and in vivo throughout intra nasal administration.

The sequence of dNP2 used in the study should be reported in the manuscript. What is the dNP2/HA2 stoichiometry?

The authors should provide the biophysical characteristics of the dNP2/siRNA complexes (size, charge, ratio...), which are essential for in vivo application.

The authors should provide the protocol use for in vivo administration? What is the meaning of ratio (molar, charge ...)?

Fig 6B : The authors propose that dNP2-HA2 forms stable complexes with siRNA, however only a poor characterization by gel retardation assay is reported. In Figure 6B, the authors suggest that dNP2/siRNA complexes are already form at 1:1 ratio, which is not really clear on gel shift assay. Complexes are mainly form at 1:25 ratio and not at 1:1 ratio. Than 1:10 ratio was selected for further evaluation can be a major concern as a large percentage of siRNA remains free in the solution. Therefore, the authors cannot only rely on dNP2-siEGFP and controls with free siRNA and free peptide are required to validate in vivo siRNA delivery experiments.

Fig 6C : What is the concentration of siRNA? what is the dNP2/siRNA ratio? The authors should provide a dNP2-siRNA dose response.

Fig 6E : What is the ratio corresponding to 2.5 μ g siRNA /70 μ g dNP2? A dNP2-siRNA dose response and controls including free siRNA and free peptide should be reported.

In Fig 6E, it is surprising that dNP2-siEGFP increased mRNA expression and dNP2-siChi3l1 has a limited effect with a rapid recovery of Chi3l1 mRNA level at day 3. Usually, as reported by several works, recovery is starting after 5 days. The authors should clarify that point.

Fig 6 (I,H) : Controls with free peptide and free siRNA should be reported. The values of 100 ng and 250 ng siRNA should correlate to the level of siRNA mediated Chi3l1 mRNA KO?

Fig 7 : The authors need to clarify the protocol of administration: every 2 days as reported in the text, or twice per day as reported in the figure. As in Fig 6, controls with free peptide or free siRNA are required.

Reviewers' comments:

Reviewer #2 (Remarks to the Author):

Overall it seems like the authors have reasonably responded to the comments from reviewers. I don't have additional comments.

Answer; Thank to Reviewer #2 of accepting our answers.

Reviewer #3 (Remarks to the Author):

Editorial note:

Reviewer#3 was unavailable to assess the revised version of your manuscript. Therefore we asked Reviewer#2 to comment on the authors rebuttal to Reviewer#3. Reviewer#2 find that the authors did a reasonable job in addressing the key issue raised by the reviewer regarding the T cell-dependency of their mechanism.

However he notes that the authors mention ongoing experiments (e.g. those mentioned on page 10 of the rebuttal) and ask to include those additional data in the revised manuscript"

Answer; We would like to thank to this reviewer #2 for considering of our answers to reviewer #3. As the reviewer asked, we completed the experiments and added it as supplementary figures including subcutaneous tumor model, TIL analysis, and CD8/Treg ratio.

1. Subcutaneous tumor model in CD4-Chi3l1 KO mice

Subcutaneous injection of B16F10 cells grow by days which did not give a significantly different tumor size results between wild type and CD4-Chi3l1 KO mice. However, IFN γ and/or TNF α expressing both CD4 and CD8 T cells were consistently increased in the tumor of CD4-Chi3l1 KO mice suggesting deletion of Chi3l1 in T cells confers more Th1 and CTL generation in vivo. Due to only limited number of T cell clone would be B16F10 specific in B6 mice, if we could utilize EG-7-OVA tumor cell transfer model in OTII background (Blood. 2012 Jun 14;119(24):5678-87.) we speculate the size of tumor could be reduced in CD4-Chi3l1 KO mice. We added the result as supplementary Fig.15. with short discussion in line 343 page 15 and method in line 489 page 20.

We also preliminary attempted subcutaneous injection of B16F10 cells that we still found significantly increased tumor-infiltrating IFN γ and/or TNF α producing CD4 and CD8 T cells although tumor size was not reduced (Supplementary Fig. 15). As a further study, we will more investigate antigen specific solid tumor model such as EG-7-OVA versus OT-II TcR transgenic background CD4-Chi3l1 KO mice.

Subcutaneous tumor growth model. 1×10^5 of B16F10 melanoma cells were s.c. injected into C57BL6/J, CD4-Chi3l1 KO mice. The size of tumor was checked for every 2 days. At day 18, mice were sacrificed, and the tumor tissue was dissected for further experiment. Tumor sizes were measured by caliper. Tumor tissues were mechanically chopped and incubated in Ca $^{2+}$ - and Mg $^{2+}$ -containing DPBS with collagenase D (1 mg/mL, Roche) for 30 min at 37°C.

Enzymatic digestions were stopped with 0.5 M EDTA solution. Digested cells were filtered by 40- μ m cell strainer, and RBCs were lysed with Ack buffer. Tumor infiltrated lymphocyte cells were enriched by Percoll gradients, and the cells were analyzed by flow cytometry.

Supplementary Figure. 15.

2. Non-TIL vs TIL

As the reviewer #3 importantly concerned tumor-infiltrating lymphocytes (TIL), we performed another set of experiments to analyze reliable number of results regarding non TIL and TIL cells in wild type and CD4-Chi31 KO mice. Each dot in non-TIL group refers one mouse while the dot in TIL group means pooling of 2 mice samples. The results suggesting there was significantly increased number of TIL cells as CD45 staining. In addition, there was increased IFN γ producing CD4 T cells in both non-TIL and TIL. More importantly, CTL/Treg ratio which

is critical for successful anti-tumor immunity (Nature. 2015 Apr 16;520(7547):373-7., Proc Natl Acad Sci U S A. 2010 Mar 2;107(9):4275-80.) was significantly increased in CD4-Chi311 KO mice suggesting reduction of lung metastasis is mediated by tumor-infiltrating effector T cells in vivo. We added this result as supplementary Fig. 4. with discussion in line 162 page 8.

Importantly we further confirmed that there were significantly increased tumor infiltrating lymphocyte (TIL) populations in the lung by CD45 staining which contains IFN γ producing T cells with increased Th1 and CTL related mRNA expression in tumor region. Although there is no significance of statistical analysis due to the limited number of samples, CD8/Treg ratio showed increased pattern in CD4-Chi311 KO mice, which is clinically critical value for successful immunotherapy^{18,19} (Supplementary Fig. 4).

Revised Supplementary Figure. 4.

Reviewer #4 (Remarks to the Author):

In the present manuscript "Inhibition of Chitinase 3-like-1 expression enhances Th1 and CTL function via the IFN γ /STAT1 axis to prevent lung metastasis", the authors demonstrate a novel immune regulatory function of Chi3l1 in T cells and suggest that Chi3l1 constitutes an effective therapeutic target to prevent pulmonary metastasis.

Although the work is of a major interest in the field of cancer immunotherapy, several points need to be addressed and protocol to be clarified.

Answer; We would like to thank to the reviewer #4 for valuable comments on our study. We pleasantly addressed all this reviewer's questions to improve our manuscript quality and clarify the protocols.

Comments

The authors previously reported a new cell penetrating peptide dNP2 isolated from the human novel LZAP-binding protein. The tandem peptide dNP2 has been successfully applied to deliver an immune regulatory protein into T cells and in vivo. In the present work, they applied dNP2 peptide to improve the delivery of siRNA targeting Chi3l1 into both cultured cells and in vivo throughout intra nasal administration.

The sequence of dNP2 used in the study should be reported in the manuscript. What is the dNP2/HA2 stoichiometry?

Answer; The strategy we combined dNP2 with HA2 is for improving endosome escape (Nat Med. 2004 Mar;10(3):310-5.) of dNP2-siChi3l1 complex. we asked peptide synthesis service to Genscript to make dNP2-HA2 with high quality (>95%). The full sequence of dNP2-HA2 peptide 'KIKKVKKKGRKGSKIKKVKKKGRK-GLFGAIAGFIENGWEGMIDG' was now added to the Method section with short explanation in line 460 page 19.

dNP2-siRNA complex formation. To determine optimal conditions for forming cell-permeable peptide-siRNA complexes, EGFP or Chi3l1 siRNA (Bioneer) was mixed with dNP2-HA2 peptide (Genscript, KIKKVKKKGRKGSKIKKVKKKGRK-GLFGAIAGFIENGWEGMIDG) which is a combined sequence of a cell-penetrating peptide dNP2 and endosomal escape sequence of HA2²² at various N/P ratios (the number of amine groups in dNP2-HA2 peptide to the number of phosphate groups in siRNA) and incubated for 30 min at RT in the dark.

The authors should provide the biophysical characteristics of the dNP2/siRNA complexes (size, charge, ratio...), which are essential for in vivo application.

Answer; Thank to the reviewer for the comment. We could measure the size of dNP2/siRNA complex and we could inform the ratio we used, while the charge of the complex should be analyzed by zeta-potential which was not available around us. The size of dNP2-HA2 peptide is 4843.93 Da. And the length of siChi3l1 and siEGFP are 21mer of duplex form siRNA. We utilized 1:25 N/P ratio which is a ratio of moles of the amine groups of cationic polymers to those of the phosphate ones of nucleotide (J Control Release. 2004 Aug 11;98(2):317-23.) for making dNP2-siRNA complex. We attempted to measure the size of dNP2-siRNA complex at

indicated N/P ratios by Nano particle size analyzer (K-one, Korea) which shows N/P ratio dependently increasing and 1:25 dNP2-siChi311 complex size would be about 300 nm. We added the result as Fig. 7c with method description and modified the text in line 186 page 9.

Revised Figure. 7c

The dNP2-siChi311 complex size was measured by Nano particle size analyzer which shows the size of the complex is increasing as the N/P ratio (Fig. 7c). The molecular complex of 1:25 indicates around 300 nm.

The authors should provide the protocol use for in vivo administration? What is the meaning of ratio (molar, charge ...)?

Answer; Thanks to the reviewer as pointing the missing description of detail protocols. For in vivo administration, the dNP2-siRNA complex was treated to the mice by intranasal instillation. Briefly, the mice were anesthetized by zoletil+rompun, and 5 μ L of dNP2-siRNA complex was injected through mouse nostril. We administrated 2.5 μ g of siRNA + 70 μ g of dNP2-HA2 peptide complex twice per day, for every other day until day 14.

The ratio we have mentioned is N/P ratio which means ratios of moles of the amine groups of cationic polymers to those of the phosphate ones of nucleotide (J Control Release. 2004 Aug 11;98(2):317-23.). We described the protocol more in detail in method section for better understanding about in vivo administration and the N/P ratio in line 516 page 21, and line 460 page 19.

In vivo dNP2-siRNA transfection. 1.25 or 2.5 μ g of Chi311 siRNA (Bioneer) or EGFP siRNA (Bioneer) was incubated with 35 or 70 μ g dNP2-HA2 peptide for 30 min at RT, in the dark. Free siRNA or Peptide-siRNA complexes were intranasally administered to mice, which were sacrificed every 24 hours for 3 days. Lung tissue was disrupted in T-PER Tissue Protein Extraction Reagent (Thermo Fisher Scientific) for protein lysates or in TRIzol for RNA extraction. Chi311 protein in the lung was measured by Chi311 ELISA kit (R&D Systems). In the lung metastasis model, 5×10^5 B16F10 cells were intravenously injected to WT mice, and then 2.5 μ g of siRNA + 70 μ g of dNP2-HA2 peptide complex was intranasally administered to anesthetized mice with zoletil+Rompun for twice per day with 10 hours interval for every other day until day 14. At 14 days after melanoma injection, mice were sacrificed and analyzed.

dNP2-siRNA complex formation. To determine optimal conditions for forming cell-permeable peptide-siRNA complexes, EGFP or Chi311 siRNA (Bioneer) was mixed with dNP2-HA2 peptide (Genscript, KIKKVKKKGRKGSKIKKVKKKGRK-GLFGAIAGFIENGWEGMIDG) which is a combined sequence of a cell-penetrating peptide dNP2 and endosomal escape sequence of HA2²² at various N/P ratios (the number of amine groups in dNP2-HA2 peptide to the number of phosphate groups in siRNA) and incubated for 30 min at RT in the dark.

Fig 6B : The authors propose that dNP2-HA2 forms stable complexes with siRNA, however only a poor characterization by gel retardation assay is reported. In Figure 6B, the authors suggest that dNP2/siRNA complexes are already form at 1:1 ratio, which is not really clear on gel shift assay. Complexes are mainly form at 1:25 ratio and not at 1:1 ratio. Than 1:10 ratio was selected for further evaluation can be a major concern as a large percentage of siRNA remains free in the solution. Therefore, the authors cannot only rely on dNP2-siEGFP and controls with free siRNA and free peptide are required to validate in vivo siRNA delivery experiments.

Answer; We truly thank to the reviewer as pointing inappropriate description which was our mistake. As the reviewer's implication on the gel retardation data, we absolutely agree that it makes complexes from 1:10 and mainly at 1:25 ratio. We utilized 1:25 ratio for further most of other experiments not 1:10 ratio. To clarify this, we added the ratio in figure image, legend, and main text in line 183 page 9. Thus, we do not think 1:25 ratio treatment experiment would have an issue of significant amount of free siRNA remained. However, we double checked the issue as follows by setting up the experiments.

30 min incubation at room temperature of the complex was analyzed by gel retardation assay that dNP2-HA2 start to form complexes with siChi311 from 1:10, and mainly at 1:25 N/P ratio (Fig. 7b), thus we chose dNP2-siRNA complex with 1:25 N/P ratio for further experiments.

Fig 6C : What is the concentration of siRNA? what is the dNP2/siRNA ratio? The authors should provide a dNP2-siRNA dose response.

Answer; The amount of siRNA for HEK293T cell transfection is 250ng with the N/P ratio of 1:25. For dose response, we performed experiment with 100, 250, 500 ng siRNA that shows very nice dose dependent decrease of both mRNA expression and protein of Chi311 in the cells. As the reviewer requested, we added the result as a Fig. 7d,e and added the concentration, N/P ratio in figure legend.

Replaced Figure 7d, e

(d) Dose dependent reduction of Chi311 mRNA expression by dNP2-siChi311 complex at 1:25 N/P ratio.

Fig 6E : What is the ratio corresponding to 2.5 µg siRNA /70 µg dNP2? A dNP2-siRNA dose response and controls including free siRNA and free peptide should be reported.

Answer; The N/P ratio is 1:25 as the other experiments. We performed experiments to show dose dependency of dNP2-siChi311 effect by comparison with free siChi311 and free peptide in vivo. The result suggests that dNP2-siChi311 reduces Chi311 expression with dose dependent manner, and free siRNA alone with equivalent amount does not affect to Chi311 mRNA expression. We added these data as supplementary Fig. 7. and modified the text in line 198 page 9.

Supplementary Figure. 7

Both mRNA and protein level of Chi311 in the lung was reduced by dNP2-siChi311 in dose dependent manner (Supplementary Fig. 7a,b), however there was no significant effect observed by free siChi311 or free dNP2-HA2 peptide (Supplementary Fig. 7c) suggesting dNP2-HA2 peptide truly enables siChi311 delivery into lungs via intranasal route and knockdown target gene expression.

In Fig 6E, it is surprising that dNP2-siEGFP increased mRNA expression and dNP2-siChi311 has a limited effect with a rapid recovery of Chi311 mRNA level at day 3. Usually, as reported by several works, recovery is starting after 5 days. The authors should clarify that point.

Answer; Frankly, it would be simple misinterpretation of dNP2-siEGFP has significantly increased Chi311 mRNA level compare to day 0 due to the * indication of statistical analysis. We moved * mark to dNP2-siChi311 treated group to avoid confusion in Fig. 7f and h. Any day of dNP2-siEGFP group is not statistically significant between them. In Figure legend, we described statistical significance of dNP2-siChi311 treated group was analyzed to dNP2-siEGFP treated group on each day in line 635 page 31.

Revised Figure. 7f, h

Statistical significance of dNP2-siChi311 treated group was analyzed to dNP2-siEGFP treated group on each day.

TAT-peptide was attempted to deliver luciferase targeting siRNA delivery intranasally Dr. SF. Dowdy group who is a leading scientist in the field published in Nature Biotechnology 2009 (Nat Biotechnol. 2009 Jun;27(6):567-71.). As their result, luciferase expression level was started to be recovered from day 2-3 which would be similar result of ours.

Fig 6 (I,H) : Controls with free peptide and free siRNA should be reported. The values of 100 ng and 250 ng siRNA should correlate to the level of siRNA mediated Chi311 mRNA KO?

Answer; Due to dNP2-siEGFP is better control including the concept of free peptide effect, as the reviewer concerned, we performed experiment with free siRNA treatment to make sure dNP2-HA2 is required. 100 ng or 250 ng of free siRNA could not affect IFN γ production by T cells however dNP2-siChi311 250 ng consistently enhanced IFN γ and TFN α producing CD4 T cells. We added this result as Supplementary Fig. 8 to discuss free siRNA effect with emphasis of dNP2 for delivering siRNA into T cells in line 207 page 10.

Supplementary Figure. 8

We confirmed there was no effect of equivalent dose of free siChi31 treatment in Th1 differentiation (Supplementary Fig. 8) suggesting critical role of dNP2-HA2 for delivery of siRNA into T cells.

As the reviewer also concerned about Chi31 expression levels in 100 and 250 ng of siRNA treatment in T cells, we confirmed that Chi31 mRNA expression was significantly reduced especially by 250 ng dNP2-Chi31 complex treatment with increased significant level of IFN γ production We added these results as Fig.7k with short description in line 206 page 10.

Revised Figure. 7k

The increase of IFN γ mRNA level by dNP2-siChi31 has correlation with significantly decrease of Chi31 mRNA level in Th1 cells (Fig. 7k).

Fig 7 : The authors need to clarify the protocol of administration: every 2 days as reported in the text, or twice per day as reported in the figure. As in Fig 6, controls with free peptide or free siRNA are required.

Answer; Thank to the reviewer regarding required clarification about protocol description. We gave the siRNA complex twice per day which was also done every other day until day 14. Thus, 14 times treatment as total per mouse. We modified the description precisely in line 521 page 21.

In the lung metastasis model, 5×10^5 B16F10 cells were intravenously injected to WT mice, and then 2.5 μ g of siRNA + 70 μ g of dNP2-HA2 peptide complex was intranasally administered

to anesthetized mice with zoletil+Rompun for twice per day with 10 hours interval for every other day until day 14.

As the reviewer suggested, we attempted B16F10 lung metastasis model with intranasal free siRNA treatment suggesting there was no significant effect of free siRNA in reducing lung metastasis found by dNP2-complex. We added these results as Supplementary Fig. 10 with short description in the text in line 233 page 11.

Supplementary Fig. 10

Free siChi311 treatment without dNP2-HA2 peptide does not effect on melanoma lung metastasis and cytokine productions by T cells (Supplementary Fig. 10) suggesting dNP2-HA2 complex is critical for inhibition of metastasis.

Reviewer #2 (Remarks to the Author):

I believe the authors have satisfactorily responded to the reviewer comments. I do not have additional comments.

Reviewer #4 (Remarks to the Author):

In this new version, the authors have answered most of my comments.
The manuscript is now acceptable for publication